# Uncertainty-Based Experience Replay
# for Task-Agnostic Continual Reinforcement Learning

**Adrian Remonda**                                                            *aremonda@know-center.at*
*Graz University of Technology and Know-Center GmbH*

**Cole Terrell**                                                              *cterrell@know-center.at*
*Graz University of Technology and Know-Center GmbH*

**Eduardo Veas**                                                             *eveas@know-center.at*
*Graz University of Technology and Know-Center GmbH*

**Marc Masana**                                                          *marc.masana@icg.tugraz.at*
*Graz University of Technology and SAL Dependable Embedded Systems*

**Reviewed on OpenReview:** <https://openreview.net/forum?id=WxHTSPS2pi>

## Abstract

Model-based reinforcement learning uses a learned dynamics model to imagine actions and select those with the best expected outcomes. An experience replay buffer collects the outcomes of all actions executed in the environment, which is then used to iteratively train the dynamics model. However, as the complexity and scale of tasks increase, training times and memory requirements can grow drastically without necessarily retaining useful experiences. Continual learning proposes a more realistic scenario where tasks are learned in sequence, and the replay buffer can help mitigate catastrophic forgetting. However, it is not realistic to expect the buffer to infinitely grow as the sequence advances. Furthermore, storing every single experience executed in the environment does not necessarily provide a more accurate model. We argue that the replay buffer needs to have the minimal necessary size to retain relevant experiences that cover both common and rare states. Therefore, we propose using an uncertainty-based replay buffer filtering to enable an effective implementation of continual learning agents using model-based reinforcement learning. We show that the combination of the proposed strategies leads to reduced training times, smaller replay buffer size, and less catastrophic forgetting, all while maintaining performance.

## 1 Introduction

Model-Based Reinforcement Learning (MBRL) has gained popularity due to its tendency to have a lower sample complexity compared to model-free algorithms (Lillicrap et al., 2015; Haarnoja et al., 2018). MBRL agents function by building a model of the environment in order to predict trajectories of future states based on *imagined* actions (Hafner et al., 2019). An MBRL agent maintains an extensive history of its experiences, its actions in response to them, and their resulting reward in an experience replay buffer. This stored information is used to train a dynamics model that iteratively predicts the outcomes of the imagined actions into a trajectory of future states. At each time step, the agent executes only the first action in the trajectory and then the model re-imagines a new trajectory given that result (Nagabandi et al., 2018; Rao, 2010; Chua et al., 2018).

By leveraging internal models to anticipate and plan, MBRL fosters dynamic adaptation to changing environments (Nagabandi et al., 2019). This adaptability makes MBRL a promising approach for continual learning scenarios. For many real-world applications, tasks are presented in a sequence of arbitrary length, accruing repetitive experiences which need to be learned efficiently. In most cases when the sequence is long,

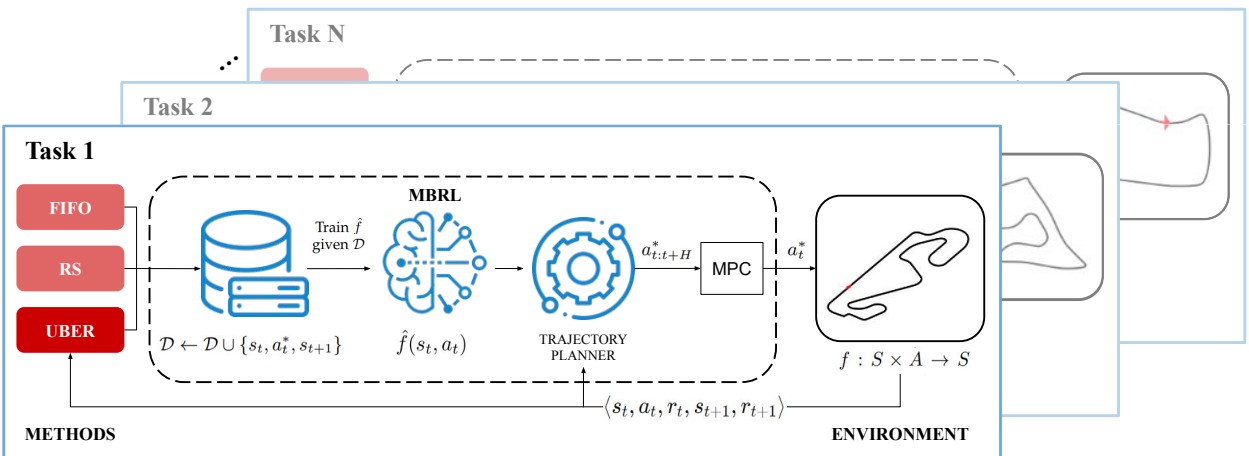

Figure 1: **Workflow.** The blue boxes showcase a single-task MBRL setting (Sec. 3, 4), and the task sequence describes a continual reinforcement learning setting (Sec. 6, 7). Highlighted in red, our method (UBER, Sec. 5) filters data added to the buffer.

it becomes unfeasible or inefficient to store all experiences and train the model on all the multiple tasks at once. Therefore, continual learning (CL) proposes to learn the tasks one at a time as they are presented to the agent (Mnih et al., 2013; Normandin et al., 2021). However, when learning a single task without any further help, performance in previous tasks decreases, which is known as *catastrophic forgetting* (McCloskey & Cohen, 1989; Kirkpatrick et al., 2017). A way to mitigate catastrophic forgetting is the usage of an experience replay buffer, which retains experiences from previous tasks used to enhance the training of the model and avoid forgetting previously learned tasks (Masana et al., 2022; Normandin et al., 2021).

Furthermore, extending training becomes a concern when accumulating similar or repetitive experiences in an unlimited replay buffer. Schaul et al. (2016) showed that transitions in a replay buffer can be more or less surprising, redundant, or task-relevant, potentially leading to a buffer inundated with redundant information which consequently under-represents other important states. Conversely, a buffer that is too small will be unlikely to retain enough relevant experiences, leading to minimal or no impact on the training of the model (Isele & Cosgun, 2018). Ideally, the size of the buffer should be the minimal amount required to capture sufficient detail for all relevant states to be learned (Zhang & Sutton, 2017). Note that, in the continual learning setting, knowing a priori all relevant states is unfeasible without extensive exploration or having access to future knowledge.

We argue that these problems can be subverted by employing a strategy that avoids retaining experiences in the buffer which the model has already sufficiently mastered. Therefore, in this work we propose a method, depicted in Figure 1, that determines what gets admitted into the replay buffer in the first place, as measured by the uncertainty of the model when learning them. Only those experiences which provide value to the model are added, thus also limiting the size of the buffer to the minimal amount needed to avoid forgetting previously learned tasks. We further evaluate our method in a proposed continual learning setting that processes a series of tasks sequentially and aims to mitigate forgetting. Therefore, our contributions are:

- Development of a method for estimating model uncertainty when predicting imagined trajectories.

- A novel approach to retain relevant experiences in the replay buffer.

- Introduction of two racing-related environments with multiple circuits for sequential task learning.

- Evaluation of generalization and catastrophic forgetting in a continual learning setting.

The adoption of these strategies enables the MBRL model to self-manage the buffer size, including under the CL setting. We consider these strategies to be critical as a starting point toward the implementation of effective and stable continual learning agents.

The remaining of this manuscript is divided as follows. Details about related work are presented in Section 2. After, this manuscript is divided into two main parts: single task MBRL and our proposed method to filter the experiences retained in the replay buffer, and the expansion towards continual learning settings. First, Section 3 provides background on MBRL methods, followed by a motivation experiment which highlights the importance of the replay buffer size in Section 4, and our proposed uncertainty-based filtering in Section 5. Second, we expand MBRL adaptation towards CL settings in Section 6, and provide an experimental analysis in Section 7. Finally, we provide discussion of the limitations and conclusions in Section 8.

## 2 Related Work

**Model-based Reinforcement Learning.** MBRL has been applied in real-world control tasks, such as robotics (Zhu et al., 2020). Compared to model-free approaches, MBRL tends to be more sample-efficient (Deisenroth et al., 2013). MBRL can be grouped into four main categories (Zhu et al., 2020). *Dyna*-style algorithms optimize policies using samples from a learned world model (Sutton, 1990). *Model-augmented value expansion* methods, such as MVE (Oh et al., 2017), use model-based rollouts to enhance targets for model-free temporal difference updates. *Analytic value gradients* can be used when a differentiable world model is available, which adjust the policy through gradients (e.g. using Gaussian processes for the dynamics model (Deisenroth & Rasmussen, 2011)). *Model predictive control* (MPC) and shooting methods use planning to select actions, but have the drawback of being computationally intensive (Rao, 2010; Chua et al., 2018). The present work belongs to the last group.

Neural networks efficiently reduce sample complexity for problems with high-dimensional non-linear dynamics (Nagabandi et al., 2018). MBRL approaches need to induce potential actions which will be evaluated with a dynamics model to choose those with best reward. Random shooting MPC methods artificially generate large number of actions and can be used to select optimal actions (Camacho et al., 2004). Neural networks are a suitable alternative to families of equations used to model the environment dynamics in MBRL (Williams et al., 2017), although they tend to make overconfident incorrect predictions. Thus, quantifying predictive uncertainty becomes crucial. Ensembles of probabilistic networks proved a good alternative to their Bayesian counterparts in determining predictive uncertainty (Lakshminarayanan et al., 2016). Furthermore, an extensive analysis about the types of model that better estimate uncertainty in the MBRL setting favored ensembles of probabilistic networks (Chua et al., 2018). The authors identified two types of uncertainty: aleatoric (inherent to the process) and epistemic (resulting from datasets with too few data points). Combining uncertainty aware probabilistic ensembles in the trajectory sampling of the MPC with a cross entropy controller demonstrated asymptotic performance comparable to SAC but with sample efficient convergence. The MPC, however, is still computationally expensive (Chua et al., 2018; Zhu et al., 2020). Quantifying predictive uncertainty provides a measure of confidence in an imagined trajectory. Remonda et al. (2021) utilized this concept to avoid unnecessary replanning by relying on sequences of actions the model is confident in, thereby reducing computations. Similarly, our approach aims to determine reliable predictions from the dynamics model in relation to the imagined actions, but as a foundation for managing the growth of the experience replay buffer.

**Use of Experience Replay in MBRL.** While an uncertainty-aware dynamics model helps to mitigate the risks of prediction overconfidence, other challenges remain, such as the shifting of the state distribution as the model trains. Experience replay was introduced by Lin (1992), and has been further improved upon. One variation of this is prioritized experience replay (Schaul et al., 2016), which aims to make learning more efficient by prioritizing transitions that are more relevant for learning, rather than sampling transitions uniformly from the replay buffer, as typically done in reinforcement learning. This method improves how the model samples experiences from the already-filled replay buffer, but does not address how the replay buffer is filled in the first place. Additionally, neither work addresses the importance of the size of the replay buffer as a hyperparameter (Zhang & Sutton, 2017). Our approach limits the replay buffer by only adding experiences that should improve future prediction capacity and keeps training time bounded to a minimum.

**Task-Agnostic Continual Learning.** We assume that agents can train on a sequence of relatively similar tasks of arbitrary length. Specifically, we aim for continuous task-agnostic reinforcement learning (Normandin et al., 2021), where task boundaries are not observed and transitions may occur gradually (Zeno et al., 2021). However, we specifically focus on the incremental case where the continual learning system learns each task from a sequence without access to previous tasks before receiving a new one. Xie & Finn (2021) develop a method that exploits data collected from previous tasks using importance sampling, although with the requirement of the agent knowing when tasks change. In our case, the task identifier is never available during training and the model has no explicit information about task transitions. In such context, an incremental learner can be seen as an autonomous agent learning over an endless stream of tasks, where the agent has to: i) continually adapt in a non-stationary environment, ii) retain memories which are useful, iii) manage compute and memory resources over a long period of time (Khetarpal et al., 2020; Thrun, 1994). Our proposed approach satisfies these three requirements. Ammar et al. (2015) focus on agents that acquire knowledge incrementally by learning multiple tasks consecutively over their lifetime. Their approach rapidly learns high performance safe control policies based on previously learned knowledge and safety constraints on each task, accumulating knowledge over multiple consecutive tasks to optimize overall performance. Knowledge is shared via a latent basis that captures reusable components of the learned policies. The latent basis is then updated with newly acquired knowledge. This results in an accelerated learning of new task and an improvement in the performance of existing models without retraining on their respective tasks.

Several works propose task-agnostic methods that do not require task information to perform continual reinforcement learning. They propose strategies to address the challenge of deciding which experiences to discard from a filled experience replay, a departure from the conventional First-In-First-Out (FIFO) approach commonly adopted. Isele & Cosgun (2018) augmented the standard FIFO buffer by selectively storing experiences over four retaining strategies: favoring surprise, prioritizing high rewards, aligning with the global training distribution (reservoir sampling (Vitter, 1985)), and ensuring broad coverage of the state space. They found that reservoir sampling effectively prevents catastrophic forgetting. Rolnick et al. (2019) proposed the CLEAR method. CLEAR, utilizes experience replay buffers to prevent forgetting. The method involves actor-critic training, integrating both new and replayed experiences. This approach adopts distributed training based on the Importance Weighted Actor-Learner Architecture (Espeholt et al., 2018). A single learning network receives both new and replayed experiences from multiple acting networks, which have their weights updated asynchronously to align with the learning network. Training is done using the V-Trace off-policy learning algorithm, applying truncated importance weights to adjust for off-policy distribution shifts. Kessler et al. (2023) demonstrated that MBRL is appropriate for the continual learning setting, evaluating strategies both for adding to and sampling from the buffer. Their findings indicate that reservoir sampling facilitates a balanced representation of experiences in the replay buffer, which in turn helps to mitigate forgetting. However, these works address the problem of sampling and discarding data from a replay buffer. In contrast, our focus is on determining what gets admitted into the replay buffer in the first place, which does not require setting a replay buffer size in advance.

## 3 Model Based Reinforcement Learning

### 3.1 Preliminaries

We consider unknown stochastic dynamical systems which we formulate as a finite-horizon Markov Decision Process (Bellman, 1957). At each time $t$, the agent is at a state $s_t \in S$, executes an action $a_t \in A$, and receives from the environment both a reward $r_t = r(s_t, a_t)$ and a new state $s_{t+1}$ according to a transition function $f : S \times A \to S$. When training a reinforcement learning policy, the goal is to maximize the accumulated reward obtained from the environment. This can be adapted to the infinite horizon setting by maximizing the sum of discounted rewards $R_t = \sum_{i=t}^{\infty} \gamma^{(i-t)} r(s_i, a_i)$, where $\gamma \in [0, 1]$.

Instead, the random shooting MPC family of MBRL algorithms artificially generates a huge amount of potential future actions, given a current state $s_t$, to select the optimal action. MBRL attempts to learn a discrete time dynamics model $\hat{f} = (s_t, a_t)$ to predict the future state $\hat{s}_{t+\Delta_t}$ of executing action $a_t$ at state $s_t$. To reach a state into the future, the dynamics model *iteratively* evaluates sequences of actions, $a_{t:t+H} = (a_t, \ldots, a_{t+H-1})$ over a longer horizon $H$, to maximize their discounted reward $\sum_{i=t}^{t+H-1} \gamma^{(i-t)} r(s_i, a_i)$. These

sequences of actions with predicted outcomes are called imagined trajectories. The dynamics model $\hat{f}$ is an inaccurate representation of the transition function $f$ and the future is only partially observable. So, the controller executes only a single action $a_t$ in the trajectory before solving the optimization again with the updated state $s_{t+1}$. The process is formalized in Algorithm 1. The dynamics model $\hat{f}_\theta$ is learned with data $\mathcal{D}_{env}$, collected on the fly. With $\hat{f}_\theta$, the simulator starts and the controller is called to plan the best trajectory resulting in $a^*_{t:t+H}$. Only the first action of the trajectory $a^*_t$ is executed in the environment and the rest is discarded. This is repeated for $T_H$ number of steps. The data collected from the environment is added to $\mathcal{D}_{env}$ and $\hat{f}_\theta$ is trained further. The process repeats for $N$ iterations. Note that generating imagined trajectories requires subsequent inference of the dynamics model to chain predicted future states $s_{t+n}$ with future actions up to the task horizon, making it only partially parallelizable.

**Dynamics model.** We use a probabilistic model to model a probability distribution of next state given current state and an action. To be specific, we use a regression model realized using a neural network similar to Lakshminarayanan et al. (2016) and Chua et al. (2018). The last layer of the model outputs parameters of a Gaussian distribution that models the aleatoric uncertainty (the uncertainty due to the randomness of the environment). Its parameters are learned together with the parameters of the neural network. To model the epistemic uncertainty (the

---

**Algorithm 1** MBRL

1: Init $\mathcal{D}$ with one iteration of a random controller
2: **for** Iteration $i = 1$ To $N$ **do**
3:     Train $\hat{f}$ given $\mathcal{D}$
4:     **for** $t = 0$ To $T_H$ **do**
5:         $a^*_{t:t+H} \leftarrow ComputeOptimalTrajectory(s_t, \hat{f})$
6:         Execute $a^*_t$ from optimal actions $a^*_{t:t+H}$
7:         $\mathcal{D} \leftarrow \mathcal{D} \cup \{s_t, a^*_t, s_{t+1}\}$ // Record outcome
8:     **end for**
9: **end for**

---

uncertainty of the dynamics model due to generalization errors), we use ensembles with bagging where the members of the ensemble are identical and only differ in the initial weight values. Each element of the ensemble has as input the current state $s_t$ and action $a_t$ and is trained to predict the difference between $s_t$ and $s_{t+1}$, instead of directly predicting the next step. Thus the learning objective for the dynamics model becomes, $\Delta s = s_{t+1} - s_t$. $\hat{f}_\theta$ outputs the probability distribution of the future state $p_{s(t+1)}$ from which we can sample the future step and its confidence $\hat{s}, \hat{s}_\sigma = \hat{f}_\theta(s, [\mathbf{a}])$. Where the confidence $s_\sigma$ captures both, epistemic and aleatoric uncertainty.

**Trajectory Generation.** Each ensemble element outputs the parameters of a normal distribution. To generate trajectories, P particles are created from the current state, $s^p_t = s_t$, which are then propagated by: $s^p_{t+1} \sim \hat{f}_b(s^p_t, a_t)$, using a particular bootstrap element $b \in \{1, ..., B\}$. Chua et al. (2018) experimented with diverse methods to propagate particles through the ensemble. The $TS_\infty$ method delivered the best results. It refers to particles never changing the initial bootstrap element. Doing so, results in having both uncertainties separated at the

---

**Algorithm 2** Get Optimal Trajectory - Planning

1: **Input**: current state $s_{init}$, dynamics model $\hat{f}$
2: Initialize $P$ particles, $s^p_\tau$, with the initial state, $s_{init}$
3: **for** $a_{t:t+H} \sim CEM(.)$, 1 To $CEMSamples$ **do**
4:     Propagate state particles $s^p_\tau$ using TS and $\hat{f}|\{\mathcal{D}, a_{t:t+H}\}$
5:     Evaluate actions as $\sum_{\tau=t}^{t+H} \frac{1}{P} \sum p = 1^P r(s^p_\tau, a_\tau)$
6:     Update CEM(.) distribution
7: **end for**
8: **return** $a^*_{t:t+H}$

---

end of the trajectory. Specifically, aleatoric state variance is the average variance of particles of same bootstrap, whilst epistemic state variance is the variance of the average of particles of same bootstrap indexes. We use also $TS_\infty$.

**Planning.** To select the best course of action leading to $s_H$, MBRL generates a large number of trajectories $K$ and evaluates them in terms of reward. To find the actions that maximize reward, we used the cross entropy method (CEM) (Botev et al., 2013), an algorithm for solving optimization problems based on cross-entropy minimization. CEM gradually changes the sampling distribution of the random search so that the rare-event is more likely to occur and estimates a sequence of sampling distributions that converges to a distribution with probability mass concentrated in a region of near-optimal solutions. Algorithm 2 shows the use of CEM to get the optimal sequence of actions $a^*_{t:t+H}$

**Reliable imagination.** Random shooting MBRL methods are computationally intensive, as they require evaluating a sequence of actions with the dynamics model for each trajectory generated. BICHO (Remonda et al., 2021) is a method that improves the runtime and training time of MBRL by reducing the computational

cost. It does this by continuing to act on an imagined trajectory until it can no longer be trusted, and then replans. The BICHO method employs a probabilistic approach to determining the reliability of a trajectory and decide when to replan. This allows for efficient exploitation of trusted trajectories while reducing computational costs. Our approach exploits the BICHO mechanism to evaluate the reliability of trajectories in order to filter out unnecessary additions to the replay buffer.

## 3.2 Baseline methods

We describe and compare the relevant baseline methods used in this work, highlighting key properties and their respective implementations (see also Table 1).

- **PETs**: Standard shooting MBRL which uses a replay buffer of infinite size. We use the implementation from the authors of PETs (Chua et al., 2018). It represents an upper bound to assess performance when retaining everything in the buffer.

- **Scratch**: We also include a baseline in which the model is trained from scratch for each individual task of the training tasks set. This means that once the model is trained on a particular task, we evaluate its performance on a separate test task set. Prior to transitioning to a new task, we discard both the model parameters and the experiences present in the replay buffer associated with that task. This baseline serves the purpose of demonstrating the model's performance without the benefit of additional experiences obtained from solving other tasks. For this baseline, we utilize an infinite buffer allocated for each task. As a result, we anticipate that this approach will yield the lowest performance scores in terms of overall performance, generalization capability, and forgetting.

- **VanillaFIFO**: To manage the continuously growing experience replay buffer, we evaluate a straight-forward *first-in, first-out* (FIFO) strategy. Once the buffer size is fixed, new experiences are added until the buffer is full. Subsequently, for each new experience added, the oldest experience in the buffer is removed to accommodate the new one. We also evaluate the impact of different buffer sizes.

- **Adaptative FIFO** (A-FIFO): Since **VanillaFIFO** retains only the latest experiences, it has a bias towards more recent tasks. Following inspiration from other online and offline incremental learning settings (Prabhu et al., 2020; Masana et al., 2022; Mai et al., 2022), we propose to reduce an adaptive version of the FIFO (adaptive-FIFO) which always retains the latest experiences from previous tasks equally distributed among all the tasks seen. Given a reply buffer of $E$ maximum experiences, at task $T$ we store a maximum of $E/T$ experiences for each task, following the FIFO strategy. When a new task is added, the necessary oldest memories from each task are removed to make room for the new task experiences. We consider that these baselines are a more representative strategy for an continual learning setting, although it does not achieve all the same properties as our proposed approach. While Adaptive-FIFO offers a more balanced strategy for a continual learning setting, it has two limitations when comparing to our method. Task number awareness: i. The method requires prior knowledge of the total number of tasks, which is not always feasible in real-world, dynamic scenarios. ii. Task Transition Awareness: Adaptive-FIFO requires an indication or marker of when a task transition occurs. This awareness is often unrealistic, as transitions in CL can be subtle or even seamless.

- **Reservoir sampling** (RS) (Isele & Cosgun, 2018; Vitter, 1985) augments *VanillaFIFO* by selectively storing experiences aligning with the global training distribution to ensure coverage of the state space. Isele & Cosgun (2018) found that RS effectively prevents catastrophic forgetting.

## 3.3 Environments

We evaluate the methods in the CartPole and Reacher environment provided by the MuJoCo (Todorov et al., 2012) physics engine. Additionally, we introduce our own proposed environments related to racing, including Masspoint and a Non-linear Bicycle model. The choice of CartPole and Reacher environments is based on their established use in RL research, allowing for meaningful comparisons with existing approaches. By

introducing our own racing-related environments, we can further assess the methods' effectiveness in more complex scenarios, which are well-suited for continual RL. (see details in the Appendix).

**CartPole (CP).** Inverted pendulum problem, which involves balancing a pole on a cart. It has movable cart that travels along a frictionless track. On top of the car a pole with one end attached to the cart, is standing upright. The goal is to keep the pole upright for as long as possible by moving the cart left or right. The agent receives a reward of 1 for every time step the pole remains upright.

**Reacher (RE).** A robotic arm, with 6 Degrees of Freedom, aiming to reach a target position in space. Given the multiple joints and their rotations. The reward function aims to minimize the distance between the end effector and the target position.

**Masspoint (MP).** An extended version of the Masspoint environment proposed by Thananjeyan et al. (2020). It is a navigation task in which a point mass moves toward a specified goal. We modified the agent's goal so that it must move as quickly as possible without deviating from a given path. The complexity of each task is determined by the geometry of the path. The reward is set to maximize speed while minimizing deviation from the path.

**Non-linear Bicycle Model (Bike).** This model captures vehicle dynamics with greater fidelity and features higher action and observation dimensions compared to MassPoint. It considers aerodynamics, tire dynamics, and rolling resistance. Additionally, we have integrated track boundaries to enhance the realism and challenge of the simulation. The control variables include steering and a combined throttle and brake system. The reward system is designed to maximize speed, and the episode ends if the car goes off track.

## 4 Motivation experiment

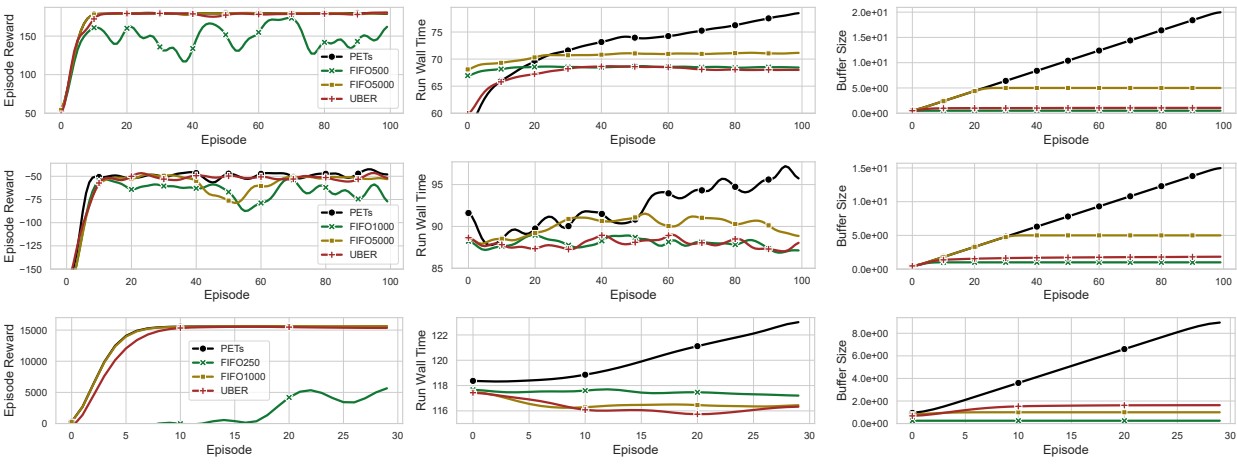

Figure 2: Performance of algorithms in (top to bottom) Cartpole, Reacher, and Masspoint. From left to right column: episode reward, time per episode (s), cumulative number of observations stored in the replay buffer. The x-axis is the number of training episodes.

In this section, we present a motivational experiment designed to analyze the growth of the replay buffer and training time when utilizing a typical MBRL method with different buffer sizes. For this experiment, we specifically employ PETs (Chua et al., 2018) on a single task. This setup intentionally mimics a controlled environment with abundant and redundant data, demonstrating how standard MBRL algorithms handle such redundancies. We aim to highlight why a standard MBRL technique falls short for continual learning. We compare an unbounded buffer size and a FIFO with different buffer sizes. We evaluate performance based on per-episode reward, per-episode wall time, and replay buffer size.

Results in the CP, RE, and Masspoint environments are depicted in Figure 2 top, middle, and bottom, respectively. Figure 2-left shows comparable reward per episode, while FIFO variants exhibit a cyclic reward, which we attribute to forgetting and regaining experiences. Results for reward in CP and RE are consistent.

FIFO rewards for Masspoint are more stable. The training time per episode (Figure 2-middle) for VanillaFIFO variants stabilizes after reaching the maximum buffer size. The wall time for PETs exhibits linear growth and takes substantially longer than FIFO to complete an episode (i.e., it takes longer to update the model as the replay buffer grows linearly).

These results shed light on the stability issues from an under-tuned replay buffer within a MBRL context, as well as the increasing training time when employing an excessively large replay buffer. These findings underscore the challenges that these issues pose to the practical applicability of MBRL in CL scenarios. Namely, the lack of memory management consequently hinders compute management and impairs learning as the agent cannot determine which experiences are useful.

**Adding experiences to the replay buffer.** We posit that it would be preferable to retain only those experiences that could not be adequately anticipated by the model during each episode in the environment. Essentially, we would like to only add to the replay buffer observations for which the model issued a poor prediction, as retaining redundant experiences will fill the buffer with unnecessary data and reduce its capacity to store meaningful, hard-to-predict observations. On the contrary, we would like to avoid filling the replay buffer or updating the model on observations that the model is good at predicting. We contend that these two elements will lead eventually to a balanced replay buffer, which will contain only relevant observations and will contribute to the objective of incremental learning. In the following section, we further describe our proposed approach.

## 5 UBER: Uncertainty-Based Experience Replay

Continual reinforcement learning requires the MBRL agent to adapt in a non-stationary environment, retaining memories that are useful whilst avoiding catastrophic forgetting, and effectively manage compute and memory resources over a long period of time (Khetarpal et al., 2020). We address these issues with our proposed method, UBER.

We propose Algorithm 3 for selecting which experiences to include in the replay buffer. In the event of an unreliable future, the algorithm replans the future trajectory and adds the current observation to the buffer. By doing so, unnecessary replanning and additions to the buffer are avoided, reducing computation time and the size of the buffer. The optimal actions $a_{t:t+H}^*$ are calculated using the $GetOptimalTrajectory$ function (as outlined in Algorithm 2) based on the current state of the environment $s_t$ and the model $\hat{f}$. The future trajectory and its uncertainty, $p_{r(t+1:t+1+H)}^*$, are then determined by using $a_{t:t+H}^*$ and $s_t$ with $\hat{f}$. The flag $unreliableModel$ is set to true when the algorithm determines that the imagined trajectory is not trustworthy. Depending on its value, further calculations and additions to the replay buffer may be avoided, reducing computation time and the size of the buffer. If $unreliableModel$ is False, the next predicted action is executed in the environment. Subsequent actions from $a_{t:t+H}^*$ are executed until the model is unreliable or the environment reaches the maximum number of steps, $T_H$. The process is repeated for the maximum number of iterations allowed per task. Hereby, the buffer stores only experiences for which the model could not predict (*imagine*) its outcome.

**Uncertainty estimation**. To determine when the predictions given by the dynamics model are still trustworthy, we utilize the BICHO method introduced by Remonda et al. (2021). BICHO evaluates the validity of the current trajectory by making probability estimates of the projected future reward, ensuring that it does not deviate significantly from the imagined future reward $p_r^*$ and that the uncertainty in the model remains low. BICHO is built under the assumption that if certain parts of the trajectory do not vary, their projected reward will align with the model's imagination with a certain level of confidence. To achieve this, after calculating a trajectory, the distribution of rewards $p_r^*$ is calculated for H steps in the future. At each step of the environment, whether a replanning step was skipped or not, a new trajectory $p_r'$ of H steps is projected, starting from the state $s_t$ provided by the environment and using actions $a_{t+i}^*$ from the imagined trajectory. We use a distance metric to find how much these two distributions change after each time step in the environment. If the change is $> \beta$ then unreliableModel is True. We can control how many steps ahead we would like to compare the two distributions. The comparison is done for LA steps ($< H$), which is a hyper parameter to tune. If the two distributions differ significantly, then the trajectory is unreliable.

---

**Algorithm 3** UBER

---

1: Initialize dynamics model $\hat{f}$ parameters; Initialize replay buffer $\mathcal{D}$ with an iteration of a random controller
2: $unreliableModel = True$ and $trainModel = False$
3: **for** Iteration $l = 1$ to $N$ **do**
4:     **if** $trainModel$ **then** Train $\hat{f}$ given $\mathcal{D}$
5:     **for** $t = 0$ to $T_H$ **do**
6:         **if** $unreliableModel$ **then**
7:             Get $a^*_{t:t+H}$ from $ComputeOptimalTrajectory$ $(s_t, \hat{f})$
8:             Get $p^*_{r(t+1:t+H)}$ given $(s_t, \hat{f}, a^*_{t:t+H})$ // Use $\hat{f}$ to predict H rewards ahead
9:             $i = 0$
10:         **else**
11:             $i \mathrel{+}= 1$
12:         **end if**
13:         Get first action $a_t$ from available optimal actions $a^*_{t:t+H}$
14:         Execute in the environment first action $a^*_t$ from remaining optimal actions $a^*_{t:t+H}$ to obtain $s_{t+1}$ and $r_{t+1}$
15:         Discard first action and keep the rest $a^*_t = a^*_{t+1:t+H}$
16:         // Uncertainty Estimation
17:         $uncertaintyScore = $ Compute $uncertaintyScore$
18:         **if** $uncertaintyScore > \beta$ **then** unreliableModel = TRUE **else** unreliableModel = False
19:         **if** $unreliableModel$ **then**
20:             Record outcome: $\mathcal{D} \leftarrow \mathcal{D} \cup \{s_t, a_t, s_{t+1}\}$
21:             // Updates on novel information
22:             **if** $new\_data\_in$ $\mathcal{D} > new\_data\_threshold * length(\mathcal{D})$ **then**
23:                 $trainModel = $ True
24:             **end if**
25:         **end if**
26:     **end for**
27: **end for**

---

**Algorithm 4** Get Uncertainty Score

---

**Input**: $i$, $p^*_{r(t+1:t+H)}$ and $LA$

1: Get $p'_{r(t+1:t+H)}$ from $ComputeTrajectoryProbs$ $(s_t, \hat{f}, a^*_{t:t+H})$
2: L = min(H, LA - i, MPD) // Calculate number of steps ahead to consider
3: $UncertaintyScore = WassersteinDistance(p'_{r(t+1:t+L)}||p^*_{r(t+i+1:t+i+L)})$
4: return $UncertaintyScore$

---

That is, if the projected reward differs from the imagined one the outcome of the actions is uncertain. We empirically evaluate different metrics and found that the Wasserstein distance (Bellemare et al., 2017) is more stable and performs better than KL-Divergence.

**Maximum prediction distance**. Even for a model that has converged, accurately predicting trajectories of great length is infeasible. Recalculations at the end of trajectories are inevitable and do not necessarily indicate the presence of new information, but rather the limitations of the successful model in a complex environment. Therefore, we exclude such recalculations from the buffer. The *maximum prediction distance* (MPD) sets a cutoff point for a trajectory and regulates the strictness of the filtering mechanism.

**Updates on novel information.** Over-training the dynamics model leads to instabilities due to overfitting. This problem is exacerbated when the replay buffer contains just the minimum essential data. If we only filter the replay buffer, continuously updating the parameters of the dynamics model will eventually lead to overfitting. Instead, our method updates the parameters of the dynamics model only when there is sufficient new information in the replay buffer.

## 5.1 Motivation experiment including UBER

The primary purpose of the proposed algorithm is for the resulting replay buffer to retain only relevant, non-redundant, experiences that will be useful for learning a single task. This experiment is intended to

show that *even when learning a single task throughout long training sessions, our method retains sufficient experiences to solve the task while curtailing buffer growth and unnecessary model updates.*

Figure 2 top shows the results obtained in CP. Figure 2-mid-right shows the size of the replay buffer during training. We observe that while the replay buffer grows in the case of PETs, the size of the buffer derived from UBER is comparably flat: the buffer resulting from UBER is 10x smaller. The training time per episode (Figure 2 mid-left) remains nearly constant and lower for UBER. FIFO variants achieve a stable wall-time after reaching the max buffer size. The wall time for PETs exhibits linear growth and takes substantially longer than both FIFO and UBER to complete an episode (i.e., it takes longer to update the model as the replay buffer grows linearly). Figure 2-left shows comparable reward per episode for PETs and UBER, whilst FIFO variants exhibit a cyclic reward, which we attribute to forgetting and regaining experiences. Results for reward in Figure 2 CP and RE (row 2) are consistent. FIFO rewards for Masspoint are more stable.

**Discarding Model Parameters.** To further demonstrate that our algorithm retains relevant samples while maintaining a minimal buffer size, we evaluate the methods by retaining and discarding model parameters. First, we train each method to convergence and save the replay buffer. then, we discard the network parameters, retrain the model using only the replay buffer from the previous step, and evaluate its performance. For each method, we report both upper and lower performance bounds. The starting point for buffer size selection is the size obtained from UBER. From there, we explore larger and smaller sizes in fixed increments. The upper bound corresponds to a buffer size beyond which further increases do not improve performance. Further, we gradually reduce the buffer size until the algorithm can no longer solve the task. This experiment aims to highlight the sensitivity of traditional MBRL methods to buffer size selection. If the experiences in the buffer are truly essential, performance should remain unchanged after discarding the model parameters.

Figure 3 illustrates the efficacy of each method when either retaining or discarding network parameters post-training. Across all environments, UBER maintains a relatively small buffer while sustaining performance, even after discarding the model parameters, retraining, and testing. The minimal performance drop after discarding weights demonstrates that UBER effectively covers the state space while maintaining a compact replay buffer with only relevant experiences. Notably, UBER generally performs on par with RS while offering the key advantage of not requiring a pre-set buffer size. This highlights an important distinction: RS and FIFO exhibit significant sensitivity to buffer size, whereas UBER mitigates this issue.

**Experiences added.** Figure 4 shows the buffer growth in Masspoint when experiences are added to the replay buffer by UBER. When the model is untrained, many experiences are added throughout the episode. However, after the model is trained, UBER stops adding experiences to the buffer as the model can predict them. As a result, new experiences are considered redundant and unnecessary for the model. The results support our claim that UBER achieves a much smaller, intelligently populated replay buffer containing only relevant information, without needing to know task boundaries or set a fixed buffer size, while still maintaining performance.

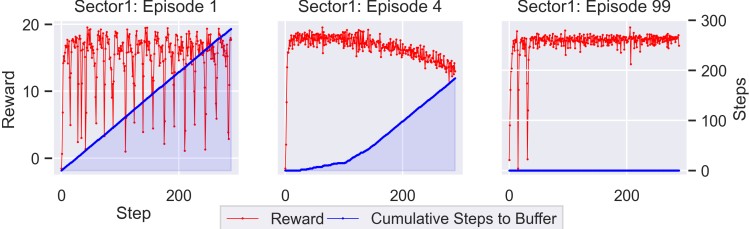

Figure 4: **Steps Added to the Replay Buffer.** Per-step reward and cumulative steps added to the replay buffer for an untrained *(left)*, partially trained *(middle)*, and fully trained model *(right)* in Masspoint. The plots show that UBER reduces redundant experiences as training progresses.

## 6 Towards continual learning

Applying MBRL to a continual learning setting is a promising avenue for research since the dynamics model could constantly improve and adapt dynamically to changes in the environment. Many real-world applications can be broken into sequences of tasks of arbitrary length, which, in some cases, support a more realistic learning regime for the agent. Although capturing the complete dynamics requires exposure to

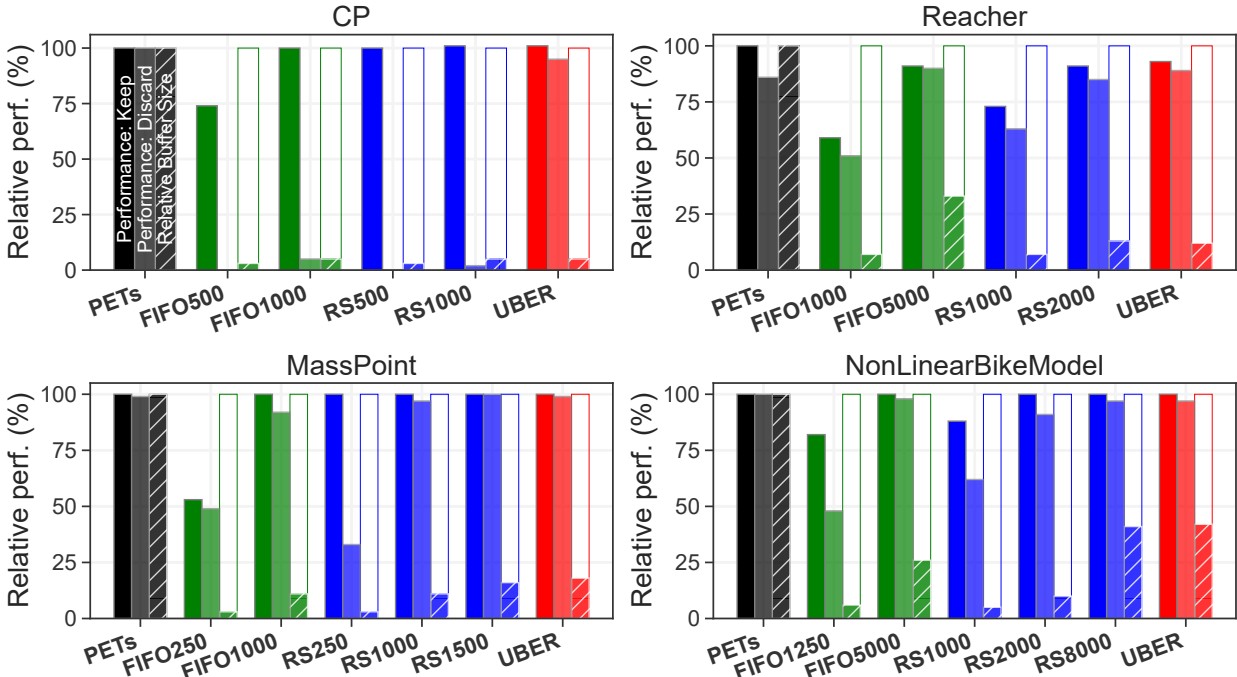

Figure 3: **Effect of Model Parameters Retention and Discard.** The left columns show results from training to convergence and subsequent testing. The middle columns show outcomes of discarding network parameters post-training while preserving experiences in the replay buffer and testing. Both left and middle columns show performance percentage relative to PETs. The right columns compare the relative size of the replay buffer to the PETs. Minimal performance drop after discarding weights shows UBER effectively covers the state space, maintaining a minimal replay buffer size with relevant experiences.

longer sessions of CL, we argue that as the model learns, the experience buffer and training time should diminish when the tasks share some similarities with previously learned ones. Arbitrarily long repetitive tasks lead to increasing redundancy in the experience replay, which constantly grows indefinitely with all collected experiences. However, we pose that for a scalable CL setting, the experience replay buffer should be limited to those experiences that properly represent the common and rare events, excluding experiences that do not provide new information to the buffer. This serves two purposes: first, to avoid an infinitely growing buffer as the sequence grows longer, and second, to provide better support and less redundancy from the experience replay.

The Cartpole and Reacher environments are typical benchmarks in RL that emphasize short-term, episodic adjustments to either maintain or reach a specific state. However, real-world settings often require continual learning with long-term adaptations, evolving contexts, and the integration of past knowledge into new tasks. We propose using the racing environment, which is diverse and versatile. It can accommodate various track geometries and shapes, and incorporates overtaking, cornering, and fuel management. This diversity provides a richer learning landscape compared to more constrained tasks like Cartpole or Reacher. By focusing on racing environments, we can gain deeper insights into the strengths and weaknesses of our algorithms, pushing the boundaries of continual learning.

## 6.1 Scenarios and task implementation

To further test our method we show that our approach helps to mitigate catastrophic forgetting. When using a fixed replay buffer size, it is important to ensure that the appropriate maximum buffer size is chosen (Zhang & Sutton, 2017). If this value is undertuned, important experiences can be jettisoned, and catastrophic forgetting can occur. The continual learning setting exacerbates this effect because, without knowing the number of tasks, the buffer size cannot be determined in advance. Thus, we experiment in

Task Agnostic Continual Reinforcement Learning where the model is not aware of tasks or task transitions. In addition, the tasks in the environments overlap, in the sense that some experiences are redundant and may occur similarly in many tasks while other experiences appear rarely or even exclusively in some tasks. This setting requires algorithms to retain releevant collection of experiences to succeed and to achieve strong performance with less data. To test the existence of these characteristics, after training on each train task, the methods are each tested on the test tasks. The model must remember what it learned by training on each sub-task and apply this knowledge to navigate a more complex, unseen task. The scenarios offer such characteristics by building on tasks aiming to traverse a path according to some imposed policy (be fast, reduce energy use).

**MassPoint tasks.** Tasks are defined as segments or as entire tracks. Tasks are defined by the geometry of the path to be followed. EASY Tasks (T01-T08) consist of straightforward and brief segments that make up the Barcelona circuit. MID tasks (T9-T11), these represent three segments of the Barcelona circuit. Note that these MID tasks partially comprise one or more EASY Task. Refer to Figures 8, 9 and 10 in the appendix. HARD Tasks (T12-T14), these tasks involve complete tracks. They are RedBullRing (T12), Barcelona (T13), and a simpler Oval (T14). In all tasks, except for T02 and T04, the objective is to achieve maximum speed. However, in tasks T02 and T04, a specific target speed is set. This demonstrates that our model is not only capable of driving at its fastest but can also maintain a designated speed.

**Bike tasks.** Similar to MassPoint, we defined a set of tasks to evaluate performance and generalization to unseen complex tasks. EASY tasks: The sub-tasks (T01, T02, T03, T04, T05, T06, T07, T08, T09) that form parts of the Barcelona (T10) circuit. HARD tasks: The three complete tracks. Barcelona (T10), Red Bull Ring (T11), and Oval (T12). Refer to Figures 11 and 12 in the appendix.

## 6.2 Evaluation metrics

When evaluating in continual learning scenarios, we need to adapt the metrics to reflect the performance based on the current performance of the model, the forgetting of previous tasks, and the generalization towards future tasks. Let $M$ be the task performance matrix where each element $M_{ij}$ represents the performance of the algorithm on test task $T_j$ after being trained on train task $T_i$. The diagonal elements of the matrix represent the immediate performance of the algorithm on a task right after training on that same task. This is denoted by $M_{ii}$, such that the algorithm is tested on task $T_i$ immediately after being trained on task $T_i$.

After the algorithm finishes training on the last training task $T_m$, its performance is evaluated across all test tasks $T_j$, for $j = 1, 2, \ldots, n$, which corresponds to the last row in the matrix. This provides the performance of the algorithm on all tasks after training on the whole sequence, by averaging over the row: $M_m = \frac{1}{n} \sum_{j=1}^{n} M_{mj}$. This is one of the main metrics used to report performance over a sequence of tasks (Normandin et al., 2021; Masana et al., 2022).

For task $T_j$, *forgetting* is reported as the difference between the performance of the algorithm when it was learned and after training on all tasks, $F_j = M_{jj} - M_{mj}$. This measures how much the performance on a task has deteriorated after the algorithm has been trained on all subsequent tasks (Chaudhry et al., 2018). We report the average forgetting across all tasks: $\frac{1}{n} \sum_{j=1}^{n} (M_{jj} - M_{mj})$.

Finally, for measuring the *generalization* capabilities of the algorithm, we can evaluate the model on unseen tasks. These would correspond to the values above the diagonal of $M$. However, we also measure the performance on unseen tasks $U_j$ (tasks not included in the training sequence) after the algorithm learns all training tasks. This measures how well the algorithm can apply the knowledge learned during training to new, unseen tasks.

# 7 Experimental analysis/results

## 7.1 Experimental Protocol

We conduct three runs for each hyperparameter configuration: varying the $\beta$ parameter for UBER, and the buffer size for the other methods. All other hyperparameters are kept consistent with the settings from

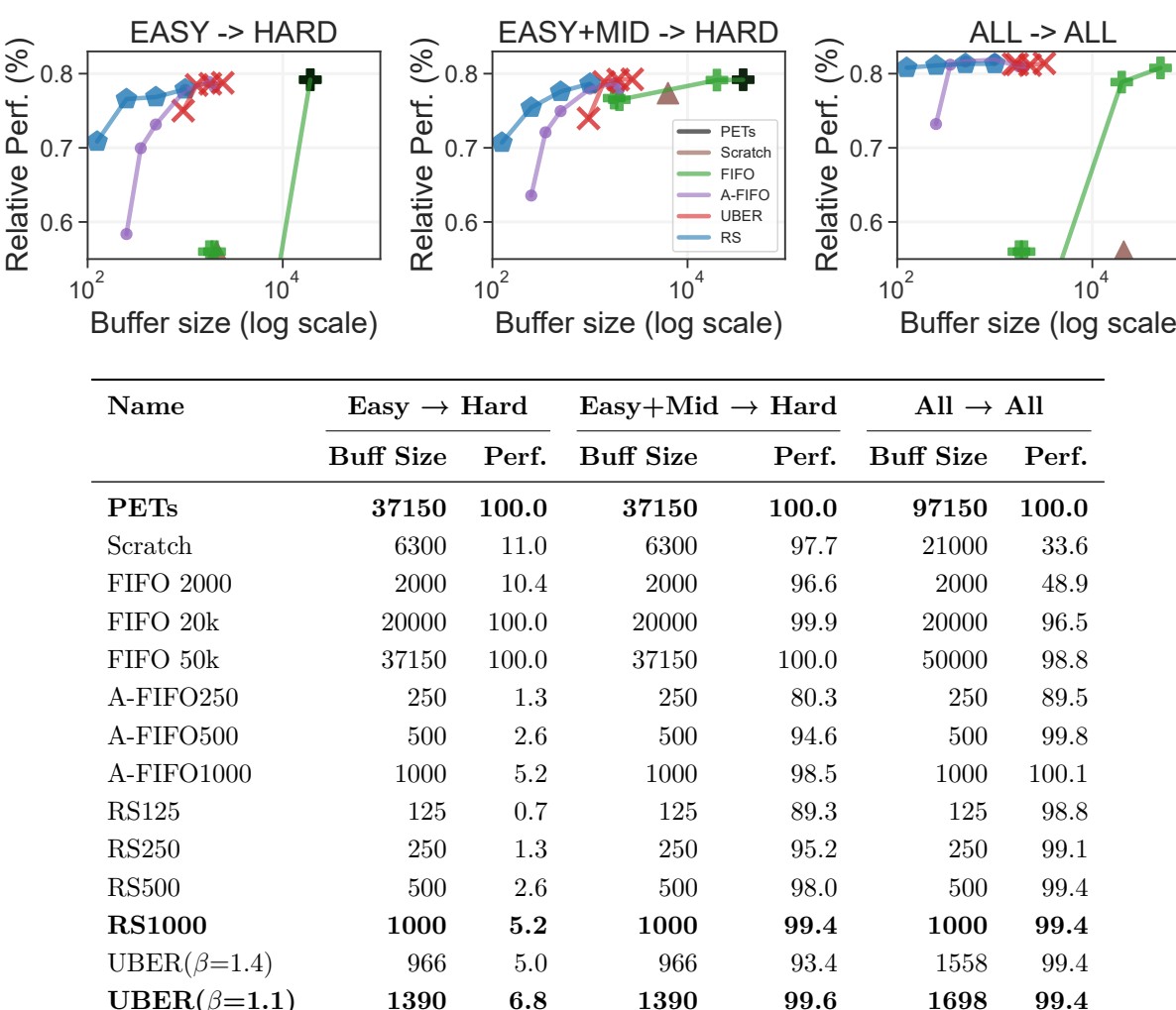

| Name | Easy → Hard | | Easy+Mid → Hard | | All → All | |
|---|---|---|---|---|---|---|
| | **Buff Size** | **Perf.** | **Buff Size** | **Perf.** | **Buff Size** | **Perf.** |
| **PETs** | **37150** | **100.0** | **37150** | **100.0** | **97150** | **100.0** |
| Scratch | 6300 | 11.0 | 6300 | 97.7 | 21000 | 33.6 |
| FIFO 2000 | 2000 | 10.4 | 2000 | 96.6 | 2000 | 48.9 |
| FIFO 20k | 20000 | 100.0 | 20000 | 99.9 | 20000 | 96.5 |
| FIFO 50k | 37150 | 100.0 | 37150 | 100.0 | 50000 | 98.8 |
| A-FIFO250 | 250 | 1.3 | 250 | 80.3 | 250 | 89.5 |
| A-FIFO500 | 500 | 2.6 | 500 | 94.6 | 500 | 99.8 |
| A-FIFO1000 | 1000 | 5.2 | 1000 | 98.5 | 1000 | 100.1 |
| RS125 | 125 | 0.7 | 125 | 89.3 | 125 | 98.8 |
| RS250 | 250 | 1.3 | 250 | 95.2 | 250 | 99.1 |
| RS500 | 500 | 2.6 | 500 | 98.0 | 500 | 99.4 |
| **RS1000** | **1000** | **5.2** | **1000** | **99.4** | **1000** | **99.4** |
| UBER($\beta$=1.4) | 966 | 5.0 | 966 | 93.4 | 1558 | 99.4 |
| **UBER($\beta$=1.1)** | **1390** | **6.8** | **1390** | **99.6** | **1698** | **99.4** |
| UBER($\beta$=0.9) | 1942 | 9.4 | 1942 | 100.0 | 2303 | 99.3 |
| UBER($\beta$=0.7) | 2712 | 12.7 | 2712 | 100.0 | 3225 | 99.5 |

Figure 5: **Generalization and Performance in Masspoint.** The top row presents performance plots showing generalization trends: *(left)* training on EASY tasks and testing on HARD tasks; *(center)* training on EASY and MID tasks and testing on HARD tasks; *(right)* training and testing on all difficulty levels. The x-axis (log scale) represents the number of experiences in the replay buffer, while the y-axis represents normalized rewards. The table below summarizes numerical results. Black cross is PETs.

the motivation experiment. Each run uses different random seeds and initial conditions. PETs is treated as a special case of VanillaFIFO, where the buffer size is effectively infinite, allowing all experiences to be retained indefinitely. For VanillaFIFO, A-FIFO, and RS, we evaluate different buffer sizes to assess the adaptive algorithm's ability to dynamically adjust and optimize memory usage based on the task at hand. For UBER, we build on our single-task experiments, keeping the same hyperparameters to validate their generalizability in more complex scenarios. However, we also explore a range of $\beta$ values to analyze its effect on performance. This experiment is set in a Task Agnostic Continual RL setting, where the model is not aware of tasks or task transitions. We utilize both the Masspoint racing environment and the more complex NonLinear Bicycle model, defining different tasks with varying levels of complexity.

## 7.2 Masspoint

We evaluate PETs, VanillaFIFO, Scratch, A-FIFO, RS, and UBER with various buffer sizes. PETs and Scratch use virtually infinite buffers. VanillaFIFO and A-FIFO are tested with different buffer sizes, and UBER with various values of $\beta$. For VanillaFIFO, buffer sizes are $BS \in 2000, 20000, 50000$. For A-FIFO, buffer sizes are $BS \in 250, 350, 500, 1000, 2000$. For UBER, $\beta$ values close to 1.1 are tested $\beta \in 0.7, 0.9, 1.1, 1.4$.

We assess performance in terms of per-episode reward, and replay buffer size. Each task is trained for 30 episodes in each task and then tested in the test tasks for a single episode. We report, generalization to unseen tasks, final performance in seen tasks, and forgetting. Each test result is normalized by dividing it by the maximum theoretical value, calculated as the top speed multiplied by the task's time steps. Normalized Value $= \frac{\text{Episode Reward}}{\text{Top Speed} \times \text{Task Time Steps}}$.

**Generalization to unseen tasks.** Figure 5-left shows the generalization from EASY to unseen HARD tasks. PETs showed excellent generalization to hard tasks. VanillaFIFO with sufficient buffer size achieved performance similar to PETs, but with a smaller buffer, it failed due to filtering out important examples. Scratch failed to generalize. A-FIFO's performance varied with changes in buffer size. This indicates that A-FIFO is sensitive to the choice of replay buffer size. RS shows a similar trend while UBER achieved same performance as PETs and A-FIFO with a significantly smaller buffer.

Figure 5-middle shows the generalization from MID to unseen HARD tasks. Similar to the previous case, PETs generalized well. Vanilla with 50k and 20k samples achieved comparable performance to PETs, while Vanilla with 2k samples underperformed. A-FIFO's behavior was consistent with the previous case. UBER achieved optimal performance with 1700 examples, which is 23% less examples than A-FIFO and just a 4% of the total examples collected by PETs.

**Same-Distribution Testing.** Figure 5 (right) shows the test performance on the same tasks used during training, including EASY, MID, and HARD tasks. VanillaFIFO's performance declines after the buffer size reaches 20k samples, indicating an increasing need for buffer size as more tasks are introduced. A-FIFO and RS achieve optimal performance with only 1k steps, demonstrating the value of information from all tasks. UBER performs consistently well across different tested hyperparameters. With a $\beta$ of 1.1, UBER's final buffer size is 37% smaller than A-FIFO's and only 1.5% of PETs' buffer size.

**Forgetting.** Figure 6 illustrates the forgetting metrics for each method and different hyperparameters. The models are trained on tasks T1 to T14. After completing each training task, the model is tested across all tasks encountered up to that point. The y-axis in each figure represents the forgetting metric, while the x-axis indicates the last task completed before the respective test. A negative number indicates forgetting, while a positive number indicates an improvement in performance.

Note that the forgetting metric assesses the extent of forgetting but does not account for the model's absolute performance on the test tasks. As a result, a model may exhibit poor overall performance yet show minimal forgetting. This explains why certain methods experience a reduction in forgetting after encountering new tasks, leading to a positive forgetting trend. This trend occurs because the model's performance improves after encountering a new task, contributing to an overall improvement.

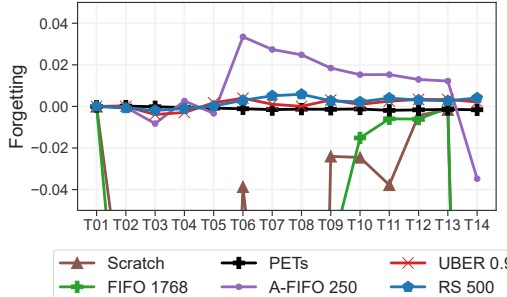

| Method | Forgetting ↓ |
|---|---|
| PETs | 0.00 |
| Scratch | 2.40 |
| FIFO1768 | 2.51 |
| A-FIFO350 | 0.20 |
| A-FIFO500 | 0.00 |
| RS250 | 0.30 |
| RS1000 | 0.02 |
| **UBER$\beta$0.9** | **0.00** |

Figure 6: **Forgetting in Masspoint.** Most methods exhibit forgetting when the buffer size is reduced. UBER maintains performance while keeping the buffer size to a minimum. The table shows the sum average forgetting for each method (↓ lower is better).

As expected, PETs has zero forgetting due to its infinitely large buffer. Conversely, Scratch has the highest forgetting rate, as it does not retain any information from previous tasks. With VanillaFIFO, reducing the buffer size limit leads to observable forgetting. In the case of A-FIFO, there is virtually no forgetting for buffer sizes larger than 500. However, reducing the buffer size limit results in an increase in forgetting. RS follows a similar trend. In the case of UBER, forgetting is virtually zero for all the tested hyperparameters except for $\beta$ 1.4. This is expected, as UBER retains experiences only when the model is not confident enough.

**Summary.** Our results in the Masspoint environment reveal that an undertuned fixed buffer size leads to poor performance and that the non-filtering algorithms hit the buffer size cap and throwing away valuable experiences, resulting in the model forgetting how to properly solve tasks that were trained early on. This is detrimental in the three aspects investigated in this experiment: generalization, forgetting and same-distribution testing. While PETs successfully excels in the evaluated aspects, it does so at the cost of having an unbounded buffer and increasing training time, making it unsuitable for the CL setting. *Scratch* training failed due to a lack of retaining important information when switching tasks. A-FIFO and RS have a good performance, but at the cost of having to tune the replay buffer size, making it dependent on the number of tasks in hand, which is not feasible in the CL setting. UBER keeps the number of examples collected to a minimum without having to tune the replay buffer size, keeping the training time at a minimum and effectively addresses catastrophic forgetting. It is the only method presented in this work suitable for the Continual Learning setting as it is the only method whose RB size does not need to be tuned depending on the number of tasks.

### 7.3   Non linear Bicycle model

We evaluate PETs, VanillaFIFO, A-FIFO, RS, and UBER, omitting Scratch due to its poor performance in this environment. For VanillaFIFO, we evaluated buffer sizes of $BS \in 25000, 50000$. For A-FIFO, we evaluated buffer sizes of $BS \in 5000, 10000, 25000, 50000$. For UBER, we evaluated different values of $\beta$ close to $1.5 \times 10^{-4}$, which gave the best results in the preliminary single-task study: $\beta \in \{2.5 \times 10^{-4}, 5.0 \times 10^{-4}, 1.5 \times 10^{-4}, 1.0 \times 10^{-4}\}$. For RS, we used buffer sizes of 5000 and 10000. We assess performance in terms of the number of steps (simulator time) to finish a task and replay buffer size. Each method is trained for 30 episodes per task and then tested on the test tasks for a single episode.

Unlike the Masspoint environment, the number of steps per episode in this environment is variable, lacking a theoretical upper bound. Therefore, we normalized the results for each task by calculating the percentage of performance drop relative to PETs (the best-performing method). We subtracted the PETs task time from the resulting task time to get the delta ($dt =$ task time $-$ PETs task time), and then normalized it using $p = 1 - \frac{dt}{\text{max\_delta}}$. Models were tested and trained on EASY, HARD and ALL tasks.

**Generalization to Unseen Complex Tasks.** Figure 7 (left) shows the generalization from EASY to unseen HARD tasks for each method with different hyperparameters. No method generalizes well to hard tasks, with each showing a significant drop in performance compared to PETs at convergence. However, increased experience leads to slightly better generalization. In the very low sample regime, RS performs worse than UBER, but at 10k samples, RS outperforms UBER. UBER's core design focuses on retaining the experiences most useful for the current task, making it less robust when the target task is out-of-distribution. This selective retention works well for within-distribution tasks but may struggle when transitioning to significantly different tasks. In contrast, RS performs better in this specific setting due to its diverse experience buffer, which is beneficial when dealing with a small number of tasks. However, RS does not scale as well to scenarios involving more tasks, since the buffer size needs to be carefully tuned. These results suggest that achieving strong generalization may require training on a broader set of diverse tasks.

**Same-Distribution Testing.** For EASY to EASY tasks (Figure 7 center left), methods with larger buffers perform better. UBER maintains reasonable performance with a very low buffer. When training on all tasks and testing on hard tasks (Figure 7 center right), and training on ALL tasks and testing on ALL tasks (Figure 7 right), VanillaFIFO performs well but requires a hand-tuned buffer. Plots show a clear increase in performance with larger buffer sizes. UBER performs similarly to RS and better than A-FIFO. PETS shows a slight performance drop when training on ALL tasks and testing on ALL tasks, indicating that redundant

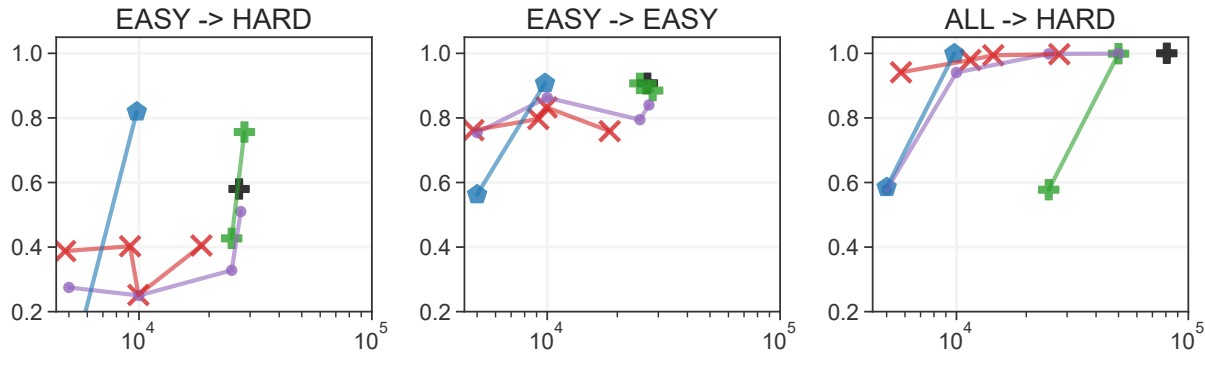

| Name | Easy → Hard | | Easy → Easy | | All → Hard | |
|---|---|---|---|---|---|---|
| | Buff Size | Perf. | Buff Size | Perf. | Buff Size | Perf. |
| PETs | 26905 | 58% | 26905 | 91% | 80752 | 100% |
| FIFO25k | 25000 | 42% | 25000 | 90% | 25000 | 58% |
| FIFO50k | 28362 | 78% | 28362 | 89% | 50000 | 100% |
| A-FIFO5k | 5000 | 27% | 5000 | 78% | 5000 | 59% |
| A-FIFO10k | 10000 | 24% | 10000 | 83% | 10000 | 95% |
| A-FIFO25k | 25000 | 35% | 25000 | 80% | 25000 | 99% |
| A-FIFO50k | 27351 | 51% | 27351 | 82% | 50000 | 99% |
| RS5000 | 5000 | 18% | 5000 | 58% | 5000 | 59% |
| **RS10000** | **10000** | **91%** | **10000** | **90%** | **10000** | **100%** |
| UBER($\beta$=0.00050) | 4819 | 39% | 4819 | 77% | 5767 | 95% |
| UBER($\beta$=0.00025) | 9156 | 40% | 9156 | 80% | 11486 | 97% |
| **UBER($\beta$=0.00015)** | **9940** | **25%** | **9940** | **82%** | **14412** | **99%** |
| UBER($\beta$=0.00010) | 18544 | 40% | 18544 | 77% | 27773 | 100% |

Figure 7: **Generalization and Performance in the Nonlinear Bicycle Environment.** The top row presents performance plots: *(left)* train on EASY tasks and test on HARD tasks (generalization); *(center)* train on EASY and test on EASY; *(right)* train on ALL and test on HARD. The x-axis (log scale) represents the number of experiences in the replay buffer, while the y-axis represents the performance drop relative to PETs. Below, the table summarizes numerical results for different buffer sizes and methods.

data is detrimental in this case. Other methods do not exhibit this issue and require smaller buffer sizes. UBER's advantage lies in not needing to set the buffer size, making it suitable for a larger number of tasks.

**Summary.** The results follow a similar trend to the MassPoint experiment in the *same-distribution* setting, but all methods perform poorly in the generalization setting, highlighting the complexity of the tasks. Our experiments show that UBER performs well independently of the number of tasks.

## 8  Discussion and Conclusion

The replay buffer enhances the stability of deep neural networks in RL and is an essential component of several algorithms. However, analyses of replay buffers are relatively scarce. Recently, research has begun to focus on analyzing the contents and strategies for managing the replay buffer in RL agents (Fedus et al., 2020), as well as in supervised learning (Aljundi et al., 2019). In this work, we contribute to this body of research by analyzing and proposing strategies to manage the growth of the replay buffer in model-based RL. In this setting, our studies show that UBER maintains a leaner and more relevant collection of experiences

| Method | Retention Strategy | Buffer Size | Buffer Specification | Task Transition Aware |
|---|---|---|---|---|
| PETs | Greedy | Fixed | Automatic (All and Keep) | No |
| PETs Scratch | Greedy | Fixed | Automatic (All and Discard) | Yes |
| FIFO | FIFO | Fixed | Manual (Hyper-parameter) | No |
| A-FIFO | Adaptive | Fixed | Manual (Hyper-parameter) | Yes |
| RS | Replacement | Fixed | Manual (Hyper-parameter) | No |
| UBER | Adaptive | Flexible | Automatic | No |

Table 1: Comparison of continual reinforcement learning properties for MBRL algorithms.

in the replay buffer than do baseline algorithms. These characteristics of the proposed algorithm, we posit, result in strong test performance with less data and greater stability.

**Comparing UBER with Existing Replay Buffer Strategies.** Table 1 provides a comparative overview of the replay buffer management strategies across various methods. Most methods employ a greedy retention strategy with a fixed buffer size. Methods like FIFO and A-FIFO utilize first-in-first-out or adaptive strategies with manual hyper-parameter tuning. In contrast, UBER distinguishes itself by using an adaptive retention strategy and a flexible buffer size with automatic management, which eliminates the need for extensive pre-tuning.

It is important to note that RS and UBER are inherently different, and each has their own strengths. UBER addresses a problem that cannot be directly addressed with RS, namely the continuous interaction and learning in the environment when we have no information about the cap size of the experience replay buffer. UBER's key advantage lies in not requiring a predefined buffer size. This flexibility is particularly beneficial in dynamic, never-ending or multi-task settings, where pre-determining the optimal buffer size is challenging. Our core contribution is a method that automatically optimizes the replay buffer without requiring preset sizes.

**Future Work.** Having managed growth, there are several aspects we would like to turn to in the future: i) identifying task boundary from the novelty of experiences, ii) managing what to forget for limited size buffers, iii) managing what to remember / refresh when a change in task is evident. This would allow to run agents for arbitrary time without having to deal with size of the buffer and would offer promising opportunities for deploying MBRL in a CL setting.

UBER could be used to prioritize entries in the replay buffer where the model was uncertain. Indeed, prioritized buffer strategies support the usage of experience once it is in the buffer, but as stated by Schaul et al. (2016), strategies for what to add and when (our work) are important open avenues for research. We did not explore our methods in environments where the tasks have interfering dynamics. But, if the dynamics change, poor predictions by the model will result in adding experiences to the replay buffer. What happens if interfering tasks occur permanently is an interesting follow up.

Expanding the experiments to high-dimensional or additional multi-task settings, such as MetaWorld, is outside the scope of the current study. UBER is implemented on top of PETs, which rely on explicitly modeling the environment dynamics and incorporating domain knowledge into the dynamics model. Implementing UBER into state-of-the-art MBRL algorithms would facilitate expansion to high-dimensional visual inputs and represent a promising research direction for future work.

**Conclusion.** We proposed strategies that comply with requirements for continual learning. Our approach retains only memories which are useful: it obtains lean and diverse replay buffers capturing both common and sporadic experiences with sufficient detail for prediction in longer learning sessions. Our approach manages compute and memory resources over longer periods: it deals with the unbounded growth of the replay buffer, its training time and instability due to catastrophic forgetting. These results offer promising opportunities for deploying MBRL in a continual learning setting.

**Acknowledgments**

Marc Masana acknowledges the support by the "University SAL Labs" initiative of Silicon Austria Labs. The Know-Center is funded within the Austrian COMET Program - Competence Centers for Excellent Technologies - under the auspices of the Austrian Federal Ministry of Transport, Innovation and Technology, the Austrian Federal Ministry of Economy, Family and Youth and by the State of Styria.

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

## Appendix

## 9    Mass point tasks - Task definition

Tasks are defined as segments of the Barcelona track (T13) or as entire tracks. Tasks are defined by the geometry of the path to be followed. In all tasks, except for T02 and T04, the objective is to achieve maximum speed. However, in tasks T02 and T04, a specific target speed is set. This demonstrates that our model is not only capable of driving at its fastest but can also maintain a designated speed. The specific definitions of the tasks are as follows: HARD Tasks (T12-T14): Displayed in Figure 8, these tasks involve complete tracks. From left to right, they are RedBullRing (T12), Barcelona (T13), and a simpler Oval (T14). MID Tasks (T9-T11): As seen in Figure 9, these represent three segments of the Barcelona circuit. EASY Tasks (T01-T08): Figure 10 depicts these tasks. They consist of straightforward and brief segments that make up the Barcelona circuit (T13).

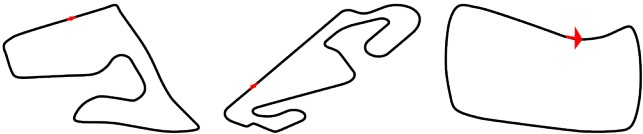

Figure 8: HARD tasks (T12-T14) for the Masspoint environment. In each figure, the x-axis and y-axis represent the x,y coordinates of the path the mass point bot should follow. The red dot denotes the starting position. Top left-to-right: RedBullRing (T12), Barcelona (T13), and a simpler Oval (T14), respectively.

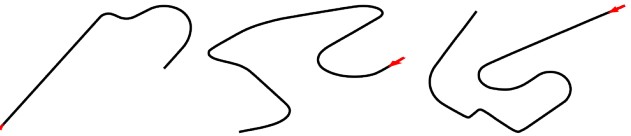

Figure 9: MID tasks (T9-T11) for the Masspoint environment. In each figure, the x-axis and y-axis represent the x,y coordinates of the path the mass point bot should follow. The red dot denotes the starting position. Left-to-right: Sector1 (T09), Sector2 (T10), and a Sector3 (T11) of the Barcelona circuit (T13).

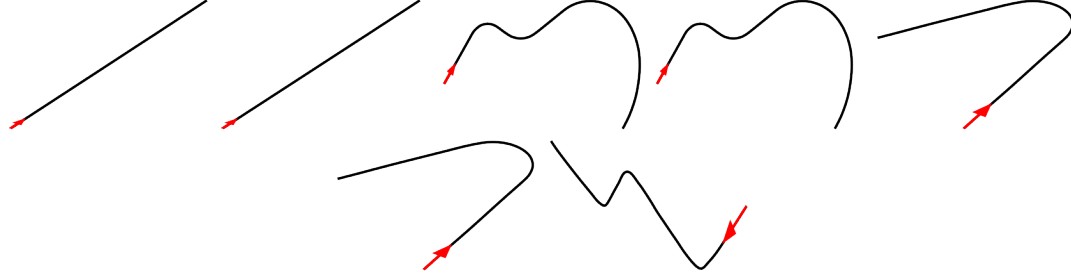

Figure 10: EASY tasks (T01-T08) for the Masspoint environment. In each figure, the x-axis and y-axis represent the x,y coordinates of the path the mass point bot should follow. The red dot denotes the starting position. These are sub-tasks taken from the Barcelona circuit (T13)

## 10    Non-linear Bicycle Model - Task definition

In the Non-linear Bicycle Model environment, we have defined a set of tasks. Figure 11 presents the three complete tracks: Barcelona (T10), RedBullRing (T11) and Oval (T12), displayed in the left, middle, and right images, respectively. Figure 12 illustrates the sub-tasks (T01, T02, T03, T04, T05, T06, T07, T08,

T09) that form parts of the Barcelona circuit. The x-axis and y-axis in each figure denote the x,y coordinates of the track path's center. The model receives data about the track borders, the objective is to ensure the car remains within the track.

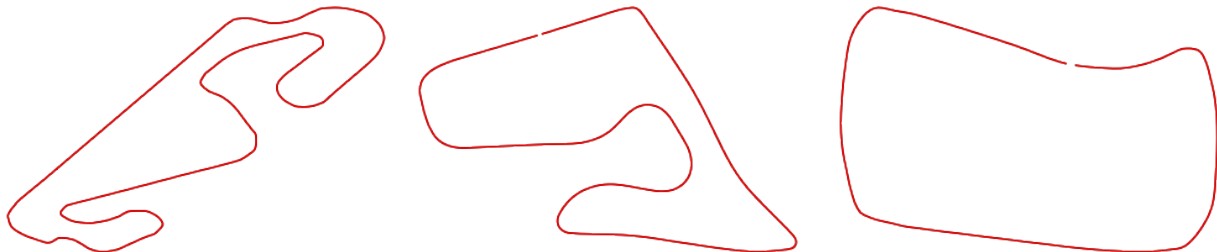

Figure 11: Tasks for the Non-linear bike model environment. The x-axis and the y-axis of each figure represents the x, y coordinates of the path to be followed by the car. Left-to-right: Barcelona (T10), RedBullRing (T11) and Oval (T12)

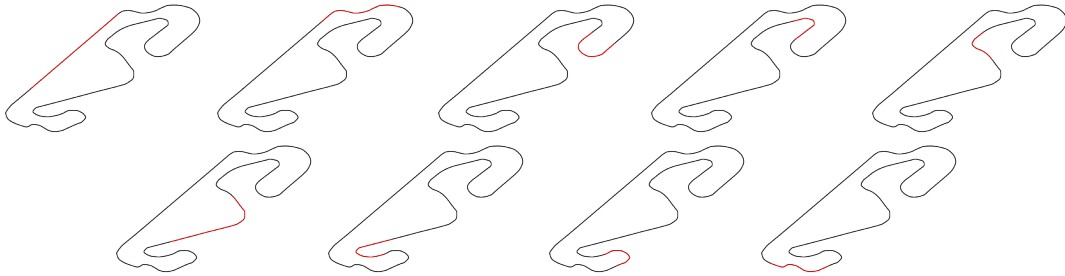

Figure 12: Sub-tasks for the Non-linear bike model environment derived from the Barcelona circuit (T10). In each figure, the x-axis and y-axis denote the x and y coordinates of the path the car should follow. Top row, left-to-right: (T01, T02, T03, T04, T05). Bottom row, left-to-right: (T06, T07, T08, T09).

## 11   Maximum Prediction Distance

A parameter of interest when using UBER is the *maximum prediction distance* (MPD). This parameter is based on the idea that even if a model has reached convergence, in certain environments, predicting very long trajectories is not feasible. Therefore, recalculations are necessary at the end of such extended trajectories. These recalculations do not necessarily indicate the arrival of new or unseen information, but instead reflect the limitations of a successful model in a complex environment. As a result, we would prefer not to add these experiences to the buffer.

The cutoff for what is considered a great length trajectory can be adjusted, allowing us to fine-tune the strictness of UBER's filtering mechanism. For example, in Ex.1 and Ex.2, we set the maximum prediction distance to 1 to apply the strictest filtering of the replay buffer.

In Figure 13, we evaluate the effect of MPD on UBER's performance in the cartpole environment, focusing on its impact on recalculation rates and replay buffer size. As shown in Figure 13, all models converge successfully, though they display slight differences in recalculation rates and buffer filtering. The strictest setting, MPD=1, results in the smallest buffer but slightly higher recalculation rates compared to models with MPD=2 and MPD=4.

These results indicate that MPD is a useful tool for adjusting the strictness of UBER's buffer filtering.

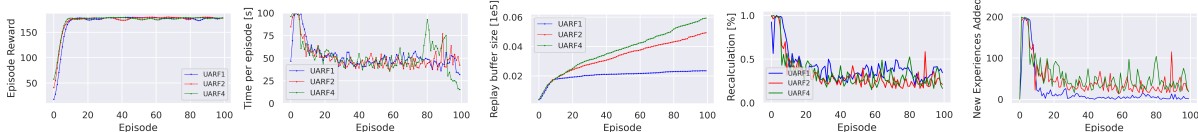

Figure 13: Performance of the examined algorithms in Cartpole using different maximum prediction distances (MPD). The blue line represents UBER with an MPD=1. The red line is UBER with an MPD=2. The green line is UBER with an MPD=4. From left to right column: episode reward, time per episode (s), cumulative number of observations stored in the replay buffer, new experiences added to the buffer per episode.

## 12 Hyperparameters

Table 2 shows the hyper parameters used to train UBER. Look-ahead refers to the number of steps ahead UBER uses to asses the quality of the imagined trajectories. $\beta$ controls the sensitivity of UBER to inform whether a trajectory is still valid or not. *New Data Train Threshold* refers to the amount of fresh data that must be added to the replay buffer before the UBER algorithm triggers the training of the dynamics model.

|  | Cartpole | Reacher | Masspoint |
|---|---|---|---|
| Look-Ahead | 1 | 1 | 1 |
| $\beta$ | 0.005 | 0.004 | 1.5 |
| Training episodes | 100 | 100 | 30/task |
| CEM population | 400 | 400 | 400 |
| CEM # elites | 40 | 40 | 40 |
| CEM # iterations | 5 | 5 | 5 |
| CEM $\alpha$ | 0.1 | 0.1 | 0.1 |
| MPD | 1 | 10 | 1 |

Table 2: Hyperparameters used for UBER implementation.

## 13 Environments

We evaluate the methods on agents in the CartPole and Reacher environment provided by the Mu-JoCo (Todorov et al., 2012) physics engine. Additionally, we introduce our own proposed environments related to racing, including Masspoint and a Non-linear Bicycle model. We utilize these environments to assess the performance of our methods. The choice of CartPole and Reacher environments is based on their established use in reinforcement learning research, allowing for meaningful comparisons with existing approaches. Introducing our own racing-related environments, such as Masspoint and Non-linear Bicycle, enables us to evaluate the methods' adaptability and effectiveness in scenarios beyond the conventional ones.

### 13.1 CartPole (CP)

An inverted pendulum problem, which involves balancing a pole on a cart. It has movable cart that travels along a frictionless track. On top of the car a pole with one end attached to the cart, is standing upright. The objective is to prevent the pole from falling over. The state variables, represented by $S \in \mathbb{R}^4$, include: the position of the cart along the track, the velocity of the cart, angle of the pole relative to vertical, the angular velocity of the pole. The action space ($\mathbb{R}^1$) is continuous, allowing the agent to apply a force $F$ to the cart, where $F$ can vary within a predefined range: $-F_{\max} \leq F \leq F_{\max}$. The goal is to keep the pole upright for as long as possible by moving the cart left or right. The episode ends after a time limit ($TaskH$) of 200 steps has been exceeded. The Trajectory horizon set for the controller ($H$), is 25. The agent receives a reward of 1 for every time step the pole remains upright.

### 13.2 Reacher (RE)

The Reacher task involves a robotic arm with 6 Degrees of Freedom (6-DoF) aiming to reach a target position in space, given the multiple joints and their rotations. State Variables: joint angles for each of the 6 joints, angular velocities for each joint, current position (x, y, z) of the end effector, target position (x, y, z). Actions: the action space is continuous, allowing the agent to apply torques to each of the 6 joints. The torque applied to each joint can vary within a predefined range: $-T_{\max} \leq T_i \leq T_{\max}$, for $i = 1, \ldots, 6$. The state space in this case is $S \in \mathbb{R}^{17}$ and the actions space is $A \in \mathbb{R}^7$. We set $TaskH$ to 150 and $H$ to 25. The reward function aims to minimize the distance between the end effector and the target position. The agent receives a reward based on the negative Euclidean distance between the current end effector position and the target.

### 13.3 Masspoint (MP)

We also included an extended version of the Masspoint environment proposed by Thananjeyan et al. (2020). Masspoint is a navigation task in which a point mass navigates to a given goal. It is a 5-dimensional $(x, y, v_x, v_y, \rho)$ state domain ($S \in \mathbb{R}^5$). Where $(x, y)$ is the position of the agent, $(v_x, v_y)$ its speed, and $\rho$ is the distance between the agent and the closest point to a given path. The agent can exert force in cardinal directions ($A \in \mathbb{R}^2$) and experiences drag coefficient $\psi$. We use $\psi = 0.6$ and included noise in the starting position. We modified the goal of the agent so that it must move as fast as possible without deviating from a given path. Each task and its complexity is then determined by the geometry of the path to be followed. The reward is calculated as $r = V(1 - |\rho|)$. Where $V$ is the speed of the agent and $\rho$ the distance to the task's path. We set $H$ to 25, and $TaskH$ depends on the task.

### 13.4 Non-linear Bicycle Model (Bike)

We introduce a new environment based on a Non-linear Bicycle Model, capturing vehicle dynamics with greater fidelity and featuring higher action and observation dimensions than MassPoint. The bicycle model simplifies a four-wheeled vehicle to a two-wheeled bicycle. This model considers aerodynamics, tire dynamics, and rolling resistance. Additionally, we have integrated track boundaries to enhance the realism and challenge of the simulation. The control variables are steering and combined throttle and brake. The state variables are the position $(x, y, \psi)$, velocity $(\dot{x}, \dot{y}, \dot{\psi})$, and acceleration $(\ddot{x}, \ddot{y}, \ddot{\psi})$, throttle, steering angle $\delta$, out-of-track, going backwards, and the distance to the reference path. Our objective is to learn the underlying dynamics model. To achieve this, we train the model to predict accelerations based on current velocities and accelerations. By integrating twice, we obtain the position and can predict trajectories. The complexity of each task depends on the geometry of the designated path. To ensure the model can generalize across different tracks, we deliberately omit the $x, y, \psi$ from the observations. Note that the model still needs to infer the $x, y, \psi$ positions, but it must do so based on the current state and the predicted accelerations. The reward aims to maximize the speed, and the episode is terminated if the car goes off track.

**Vehicle dynamics.** The state includes the position variables $x, y, \psi$, their first derivatives $\dot{x}, \dot{y}, \dot{\psi}$, their second derivatives $\ddot{x}, \ddot{y}, \ddot{\psi}$, Throttle, $\delta$, out-of-track, going-backwards, and closest-distance-to-path. $x$ and $y$ represent the coordinates of the center of mass in an inertial frame (X, Y). $\psi$ is the inertial heading. $\dot{x}$ and $\dot{y}$ are the longitudinal and lateral speeds in the body frame, respectively. $\dot{\psi}$ denotes the yaw rate. $\delta$ represents the steering wheel angle. Throttle is the combined signal of the throttle and brake pedal, representing the external longitudinal force. out-of-track and going-backwards are boolean signals, indicating when the model is off the track or moving in reverse, respectively. Closest-distance-to-path measures the nearest distance to the designated path. Our objective is to learn the underlying dynamics model. To achieve this, we train the model to predict accelerations based on current velocities and accelerations. By integrating twice, we obtain the position and can predict trajectories. We introduce noise into the initial position and velocity. The complexity of each task hinges on the geometry of the designated path.

The nonlinear continuous time equations that describe a dynamics bicycle model in an inertial frame are:

$$\ddot{x} = \dot{\psi}\dot{y} + a_x \tag{1}$$

$$\ddot{y} = -\dot{\psi}\dot{x} + a_y \tag{2}$$

$$\ddot{\psi} = \frac{F_{fy} \cdot L_f \cdot \cos(\delta) - F_{ry} \cdot L_r}{I_z} \tag{3}$$

$$\dot{X} = \dot{x}\cos\psi - \dot{y}\sin(\psi) \tag{4}$$

$$\dot{Y} = \dot{x}\sin\psi + \dot{y}\cos(\psi) \tag{5}$$

Where $a_y$ is $\frac{\sum F_y}{m}$ which are the sum of lateral forces include a simplified tire model, aerodynamics and rolling resistance (see the pseudo code). $Fc_f$ and $Fc_r$ denote the lateral tire forces at the front and rear wheels, respectively, in coordinate frames aligned with the wheels.

**Learned dynamics model.** The dynamics model aims to predict the next state given the current state (e.g. speed and accelerations) and controls (steer, throttle and brake). With a good estimate of the future state, we expect to reliably predict trajectories on a finite horizon. We train $\hat{f}_\theta$ to predict the dynamics of the vehicle, hence the accelerations of the car which can be then integrated twice to recover position.

Applying Newton's second law yields $\frac{\sum F_y}{m} = a_y$, where $a_y$ is the vehicle's inertial acceleration at the gravity center in the direction of the y axis, $m$ is the total mass of the car, and $\sum F_y$ is the summation of the lateral forces. Two terms contribute to $a_y$: acceleration due to the motion in the y axis $\dot{v}_y$, from which the position is recovered by integrating and the centripetal acceleration $v_x\dot{\psi}$ (Rajamani, 2011):

$$a_y = \frac{\sum F_y}{m} = \dot{v}_y + v_x\dot{\psi} \therefore \dot{v}_y = a_y - v_x\dot{\psi} \tag{6}$$

Similarly, equation equation 7 defines $a_x$, where $\dot{v}_y$ is the acceleration in the x axis and $v_y\ddot{\psi}$ is the centripetal acceleration:

$$a_x = \dot{v}_x - v_y\dot{\psi} \therefore \dot{v}_x = a_x + v_y\dot{\psi} \tag{7}$$

Which can then be integrated to recover the position of the vehicle. We split the state vector in kinematic state variables $\mathbf{s}_k$ and dynamics state variables $\mathbf{s}_d$, $\mathbf{s} = \begin{pmatrix} \mathbf{s}_k \\ \mathbf{s}_d \end{pmatrix}$. The kinematic part of the state is $\mathbf{s}_k = (p_x, p_y, \psi)^\intercal$ and the dynamic part is $\mathbf{s}_d = (v_x, v_y, \dot{\psi}, R)^\intercal$ and the actions $\mathbf{a}$ are given by $(S, T, B, G)^\intercal$ (Kong et al., 2015):

$$\mathbf{s}_k(t+1) = \mathbf{s}_k(t) + \mathbf{k}(\mathbf{s}(t))\Delta t \tag{8}$$

The coordinate transformation matrix $\mathbf{k}$ defined by equation 9 maps the position vector from the vehicle to the inertial frame of reference.

$$\mathbf{k}(s) = \begin{pmatrix} cos(\psi)v_x - sin(\psi)v_y \\ sin(\psi)v_x + cos(\psi)v_y \\ \dot{\psi} \end{pmatrix} \tag{9}$$

The full state space equation which we define as the *dynamics model* ($\mathbf{F}$) becomes:

$$\mathbf{F}(\mathbf{s}_t, \mathbf{a}_t) = \mathbf{s}_{t+1} = \begin{pmatrix} \mathbf{s}_k \\ \mathbf{s}_d \end{pmatrix}_t + \begin{pmatrix} \mathbf{k}(s) \\ f_\theta(\mathbf{s}_d, \mathbf{a}) \end{pmatrix}_t \Delta t \tag{10}$$

Where the dynamic part is:

$$\mathbf{s}_d(t+1) = \mathbf{s}_d(t) + f_\theta(\mathbf{s}_d(t), \mathbf{a}(t))\Delta t \tag{11}$$

We get the learning objective for the neural networks as:

$$y = \frac{\mathbf{s}_d(t+1) - \mathbf{s}_d(t)}{\Delta t} = \left(\dot{v}_x, \dot{v}_y, \ddot{\psi}\right)^{\mathsf{T}} \tag{12}$$

Thus, the neural network model is:

$$\left(\dot{v}_x, \dot{v}_y, \ddot{\psi}\right)^{\mathsf{T}}_{t+1} = f_\theta(\mathbf{s}_d(t), \mathbf{a}(t)) \tag{13}$$

### 13.5 Pseudo code

```
epsilon = 1e-6
c_a = 1.36            # Aerodynamics coefficient
c_r1 = 0.01           # Road rolling resistance
max_steer = radians(30.0)  # [rad] max steering angle
L = 2.9               # [m] Wheel base of vehicle
dt_physics = 0.1      # [s] physics sampling time
Lr = L / 2.0          # [m]
Lf = L - Lr
Cf = 1600.0 * 2.0    # N/rad      # Front tires coefficient
Cr = 1700.0 * 2.0    # N/rad      # Rear tires coefficient
Iz = 2250.0           # kg/m2      # Inertia
m = 1500.0            # kg         # mass of the vehicle

def next_state(delta, throttle):
    # scale from -1,1 to to -max steer max steer
    delta = delta * max_steer
    delta = clip(delta, -max_steer, max_steer)

    # Position in the non-intertial frame of reference
    x = x + x' * cos(psi) * dt - y' * sin(psi) * dt
    y = y + x' * sin(psi) * dt + y' * cos(psi) * dt
    psi = psi + psi' * dt
    psi = normalize_angle(psi)

    # forces
    Ffy = -Cf * arctan2(((y' + Lf * psi') / (x' + epsilon) - delta), 1.0)
    Fry = -Cr * arctan2((y' - Lr * psi') / (x' + epsilon), 1.0)
    # Aerodynamics
    F_aero = c_a * x' ** 2
    # Road rolling resistance
    R_x = c_r1 * x'
    F_load = F_aero + R_x

    # Dynamics -> to be learned by the NN
    x'' = (throttle - Ffy * sin(delta) / m - F_load / m + y' * psi')
    y'' = (Fry / m + Ffy * cos(delta) / m - x' * psi')
    psi'' = ((Ffy * Lf * cos(delta) - Fry * Lr) / Iz)

    # Velocities relative to the body frame
    x' = x' + x'' * dt
    y' = y' + y'' * dt
    psi' = psi' + psi'' * dt
    t += dt
```

