# OpenReview forum: "Uncertainty-Based Experience Replay for Task-Agnostic Continual Reinforcement Learning"
_TMLR — Accepted by TMLR_

### Review · Reviewer_d2yw · 2024-10-31

**Summary Of Contributions:**

The paper, "Uncertainty-Based Experience Replay for Task-Agnostic Continual Reinforcement Learning," introduces Uncertainty-Based Experience Replay (UBER) to manage experience retention in model-based reinforcement learning (MBRL) for continual learning (CL). UBER selectively stores experiences based on predictive uncertainty, aiming to minimize buffer size while retaining essential information for sequential task learning. This method targets catastrophic forgetting by dynamically filtering experiences before adding them to the replay buffer, which contrasts with traditional uniform sampling.

**Audience:**

Yes

**Claims And Evidence:**

Yes

**Requested Changes:**

Please see above

**Strengths And Weaknesses:**

**Strengths:**

* Efficient Memory Management: UBER focuses on storing only the most informative experiences by filtering based on predictive uncertainty, which can significantly reduce memory requirements. This is especially advantageous in continual learning, where memory constraints are often a bottleneck.

Mitigation of Catastrophic Forgetting: By prioritizing uncertain experiences, UBER helps to reinforce memories for tasks that might otherwise be forgotten over time, potentially improving long-term performance in sequential tasks. Although, the novelty surrounding might require few attributions - eg. PER [1], IER. [2], and few others.

[1] https://arxiv.org/abs/1511.05952
[2] https://arxiv.org/abs/2206.03171

* Task-Agnostic Design: UBER’s filtering mechanism does not rely on explicit task boundaries, which makes it adaptable to real-world scenarios where task shifts are gradual and not clearly defined. This aligns well with the objective of task-agnostic continual learning.

**Weaknesses:**

* Reliance on Uncertainty Estimation: Calculating predictive uncertainty accurately in complex environments can be computationally intensive. If the uncertainty estimation is unreliable, UBER could either store redundant experiences or discard valuable ones, impacting performance. Furthermore, the costs of these uncertainty estimations are unclear. For instance, it might be the case that this measure might be more simpler environments, and the costs in atari environments might hinder the speedup. Further exploration here would help better understand when this would be useful.

* Potential Overfitting: Since UBER filters experiences aggressively, there is a risk of underfitting if critical, less uncertain experiences are consistently filtered out. This could lead to over-specialization in certain areas of the task space. Could this not be a problem, when we are attempting a multi-task RL problem through CL?

* Generalizability and Scalability Concerns: UBER’s performance has yet to be demonstrated in high-dimensional or vision-based reinforcement learning tasks, where feature space complexity could challenge the efficacy of uncertainty-based filtering. I would suggest Atari or gridworld, MT50, or other larger benchmarks to help prove the scalability aspect of UBER.

* Limited Exploration: By focusing mainly on uncertain experiences, UBER might reduce exploration over time, potentially neglecting more stable experiences that could reinforce the policy and benefit long-term learning. Although such works of prioritized sampling have been presented earlier, and the exact difference other than the selective thresholding aspect is not clear. Please note, that this by itself is a useful contribution by itself, but how one manages the earlier weaknesses mentioned above is not exactly clear to me and your response would be helpful.

---

> ### Author Response · Authors · 2024-11-21
>
> We thank the reviewer for their valuable feedback. We address your comments and questions below.
>
> **Q**: Reliance on Uncertainty Estimation: Calculating predictive uncertainty accurately in complex environments can be computationally intensive. If the uncertainty estimation is unreliable, UBER could either store redundant experiences or discard valuable ones, impacting performance. Furthermore, the costs of these uncertainty estimations are unclear. For instance, it might be the case that this measure is suitable for simpler environments, and the costs in Atari environments might hinder the speedup. Further exploration here would help better understand when this would be useful.
> **A**: We assume that the uncertainty estimation is already computed by the algorithm, PETS in this case, so there is no additional overhead. Modern MBRL algorithms, such as Dreamer or TD-MPCs, can also provide uncertainty estimates (e.g., policy entropy or value function uncertainty). Our method is compatible with any MBRL algorithm that models uncertainty, making it broadly applicable.
>
> **Q**: Potential Overfitting: Since UBER filters experiences aggressively, there is a risk of underfitting if critical, less uncertain experiences are consistently filtered out. This could lead to over-specialization in certain areas of the task space. Could this not be a problem when attempting a multi-task RL problem through CL?
> **A**: From our tests, this risk is mitigated by training on longer and more diverse CL sessions (e.g., Bike and MP). The minimal forgetting observed suggests that the algorithm primarily filters redundant data. For example, in tasks with many straight segments, UBER prioritizes filtering out these redundant samples and retains more diverse experiences, such as data from corners, to keep the buffer size to a minimum.
>
> **Q**: Generalizability and Scalability Concerns: UBER’s performance has yet to be demonstrated in high-dimensional or vision-based reinforcement learning tasks, where feature space complexity could challenge the efficacy of uncertainty-based filtering. I would suggest Atari, gridworld, MT50, or other larger benchmarks to help prove the scalability aspect of UBER.
> **A**: We appreciate the suggestion. However, PETS, which we use in our current setup, does not work with image-based tasks and requires explicit modeling of the underlying dynamics. Extending our approach to high-dimensional environments is an exciting direction for future work. See also the general remark 1.
>
> **Q**: Limited Exploration: By focusing mainly on uncertain experiences, UBER might reduce exploration over time, potentially neglecting more stable experiences that could reinforce the policy and benefit long-term learning.
> **A**: The issues we observed in our method are similar to those seen in other RL algorithms. As the policy improves, exploration naturally decreases since the agent becomes more confident in its actions. However, our method can also be interpreted as a form of maximum entropy exploration, as we intentionally retain more surprising experiences in the buffer. By doing so, we maintain a level of diversity in the learning process, which helps mitigate the potential drawbacks of reduced exploration over time. We acknowledge this limitation and emphasize that our current work does not address exploration directly. Addressing exploration challenges is part of our planned future work, where we aim to integrate uncertainty filtering with strategies that enhance exploration systematically.
>
> **Q**: Although such works of prioritized sampling have been presented earlier, the exact difference—other than the selective thresholding aspect—is not clear. Please note that this by itself is a useful contribution, but how one manages the earlier weaknesses mentioned above is not exactly clear to me, and your response would be helpful.
> **A**: Our approach differs from previous methods, such as Prioritized Experience Replay, in that we focus on filtering examples before they are added to the replay buffer, rather than sampling from or pruning existing samples already in the buffer. This distinction provides a significant advantage: our method eliminates the need to predefine the size of the replay buffer.
>
> ---
>
> Please do not hesitate to let us know if you have any additional comments.

---

> > ### Comment · Reviewer_d2yw · 2024-12-20
> >
> > Thank you for the detailed response. The rebuttal answers most of my concerns. I would suggest adding these details into the final draft so that it would help other readers as well.

---

### Review · Reviewer_krR9 · 2024-11-05

**Summary Of Contributions:**

The authors propose Uncertainty-Based Experience Replay (UBER), for managing the replay buffer in model based RL. UBER leverages the model's uncertainty in its predictions to selectively add experiences to the buffer. The intuition is that experiences about which the model is uncertain are more informative and therefore more valuable for learning and avoiding forgetting. If the model can confidently predict the outcome of an action, the corresponding experience is considered redundant and not added to the buffer. This filtering mechanism allows UBER to maintain a smaller, more relevant buffer, potentially leading to faster training, reduced memory usage, and less forgetting. The authors evaluate UBER in both single-task and task-agnostic continual learning scenarios across several environments, including racing-inspired domains.

**Audience:**

Yes

**Broader Impact Concerns:**

I don't think a Broader Impact Statement is necessary.

**Claims And Evidence:**

Yes

**Requested Changes:**

- It would be useful to evaluate UBER on more multi task environments such as MetaWorld. This is the most important change that I think is needed for acceptance.
- Provide a comparison of UBER vs RS and why a practitioner might choose one over the other.
- Section 4 aims to highlight why MBRL fails in CL. But 2/3 environments (CP and RE) used are single task environments, how are they relevant when motivating UBER for continual learning?
- It might be easier to read the results in Fig 5 and 6 if they are presented as a table. It would also address the issue of values being outside the range of the y axis in those figures.

**Changes for clarity**
- In Algorithm 3 and 4:
  - Given that $a_t^*$ is discarded from the action sequence in L15, I don't think it is correct to index it by $t+i$ in L1 of Algorithm 4. $s_t$ would be the current state and you want to evaluate the log probs for the remaining actions in the sequence. Otherwise L15 should be removed from Alg 3
  - Furthermore, the projected reward $p_r^'$ in Alg 4 should be from $t$ since it is from the current time step and is recalculated every step. Only $p_r^*$ should be from $t+i+1$
  - The maximum prediction distance (MPD) is never mentioned in the algorithm. Shouldn't the prediction of actions and $p_r^*$ be only for MPD and not H? This is especially relevant since MPD=1 for 3/4 environments tested.
- The Appendix lists the value of LA for all environments as 1. This means that the uncertainty is only measured for the next step. Isn't it too short sighted for measuring the uncertainty of the model? Could the authors elaborate on why this value was chosen? I would like the value of LA be mentioned in the main text itself along with an explanation of the rationale.

**Strengths And Weaknesses:**

**Strengths**
- Simple and Intuitive: The core idea behind UBER is simple and intuitive: prioritise experiences that the model finds surprising. This simplicity makes it easy to understand and potentially integrate into existing MBRL algorithms.
- UBER addresses a relevant issue in the continual learning domain. It isn't feasible or desirable to store all experience when training an agent on a sequence of tasks, and research in filtering out redundant data is of practical importance.
- The empirical results demonstrate that UBER effectively manages the buffer size, keeping it smaller than baseline methods while maintaining performance. This efficiency is crucial for long-running MBRL agents and continual learning scenarios.
- The discard experience experiment is a good demonstration of the relevance of the their setting.
Mitigation of Forgetting: The experiments suggest that UBER can help mitigate catastrophic forgetting in continual learning, although further investigation in more complex continual learning (CL) settings is needed.

**Weaknesses**
- Limited Scope of CL Evaluation: The continual learning experiments are limited to only two environments. Evaluating UBER on more complex and diverse CL benchmarks would significantly strengthen the validity of the claims
- In some environments, RS seems to be better than UBER at achieving higher/equivalent performance with a smaller buffer size (Fig 3 & 5). What would the benefits of UBER be over RS?
- Some of implementation choices are unclear, please see the requested changes section.

---

> ### Author Response · Authors · 2024-11-21
>
> We thank the reviewer for their valuable feedback. We address your comments and questions below.
>
> ## Requested Changes
>
> **Q**: It would be useful to evaluate UBER on more multi-task environments such as MetaWorld. This is the most important change that I think is needed for acceptance.
> **A**: Currently, we are constrained by the use of PETS, which makes adding new environments non-trivial due to its requirements for explicit dynamics modeling. However, we plan to expand our work using more modern MBRL methods in the future, as outlined in the general response.
>
> **Q**: Provide a comparison of UBER vs RS and why a practitioner might choose one over the other.
> **A**: Thank you for the suggestion. We will expand the text to include a detailed comparison between UBER and RS. Table 6 already provides an initial comparison, but we plan to improve it by clarifying scenarios where each method has specific advantages.
>
> **Q**: Section 4 aims to highlight why MBRL fails in CL. But 2/3 environments (CP and RE) used are single-task environments. How are they relevant when motivating UBER for continual learning?
> **A**: In Section 4 we look into a single-task environment and the effect of redundancy in experiences when training for longer periods. We recognize that the text in Section 4 could cause confusion. We suggest a restructuring and clarification. See also the general remark 3 about the motivation experiment.
>
> **Q**: It might be easier to read the results in Fig 5 and 6 if they are presented as a table. It would also address the issue of values being outside the range of the y-axis in those figures.
> **A**: Thank you for this feedback. We will reformat these results into tables to improve clarity and ensure all values are appropriately represented.
>
> **Q**: The Appendix lists the value of LA for all environments as 1. This means that the uncertainty is only measured for the next step. Isn't it too short-sighted for measuring the uncertainty of the model? Could the authors elaborate on why this value was chosen? I would like the value of LA to be mentioned in the main text itself along with an explanation of the rationale.
> **A**: We evaluated several values for LA and MPD but found that setting both to 1 provided the most stable results across environments. We will update the main text to include this explanation and clarify this.
>
> **A**: We updated the equations and we also included the MPD hyperparameter in the algorithm.
>
> ---
>
> Please do not hesitate to let us know if you have any additional comments.

---

> ### Comment · Reviewer_krR9 · 2024-12-10
>
> Thank you for your clarifications and the general response. They have resolved my concerns with the paper. I have updated my recommendation.

---

### Review · Reviewer_beJY · 2024-11-06

**Summary Of Contributions:**

This paper addresses the challenge of managing replay buffer growth in continual model-based reinforcement learning (RL). The authors introduce "Uncertainty-Based Experience Replay" (UBER), a method that leverages the BICHO algorithm to measure model accuracy. UBER monitors the discrepancy between predicted and actual rewards in transitions to filter stored experiences effectively. By admitting only uncertain samples, UBER reduces memory use and mitigates catastrophic forgetting, enhancing efficiency in long task sequences while maintaining performance.

**Audience:**

Yes

**Broader Impact Concerns:**

I do not have broader impact concerns. This work is theoretical in nature.

**Claims And Evidence:**

No

**Requested Changes:**

1. Improve Figure Legends and Color Scheme:

* Update all figures with clear legends and consistent symbols to ensure each plot is independently interpretable. For example, symbols such as the black cross in Figure 5 should be explicitly defined in the legend.
* Use a color scheme that is color-blind accessible for better readability and accessibility. Also enhance consistency across figures.

2. Experiment Settings Demonstrating Clear Superiority of UBER over RS:
* Currently, the experimental evidence supporting UBER’s superiority over RS is limited to the CartPole (CP) results in Figure 3 and the very low data regime in Figure 7.
* I would like to see at least one additional experiment where UBER consistently outperforms RS, particularly in settings relevant to the paper’s contributions. For instance, it would be valuable to test UBER’s effectiveness in scenarios where RS might be limited by buffer size constraints or task complexity.

3. Pixel-Based Experiment:
* If feasible, add an experiment in a pixel-based setting to illustrate UBER’s advantage in managing replay buffer size. In rich visual environments, buffer constraints become more impactful, and UBER’s performance in this context would highlight its relevance to more complex, real-world applications where retaining all observations is prohibitively costly.

**Strengths And Weaknesses:**

## Overall
Strengths:
* Extensive experimentation, and the experiments are motivated.

Weaknesses:
* The results lack sufficient evidence to conclusively demonstrate UBER’s superiority over Reservoir Sampling (RS).
* All experiments are conducted in relatively simple state spaces where replay buffer size is not as critical. Testing UBER in visually rich settings would better highlight its effectiveness, as large replay buffers become significantly costlier in those environments.
* Figure legends are not comprehensive.
* Results are not well explained in the text.

## Details

### Figure 2
* Color Accessibility: The color choices are not color-blind friendly, which hinder accessibility.
* Comparison with RS: It is unclear why RS was not included as a baseline in this figure. Comparing UBER to RS here would provide a more comprehensive view of UBER’s effectiveness.

### Figure 3

* Baseline Hyperparameters: Could you provide additional context on the baseline hyperparameter selection in Figure 3? Detailing the rationale behind these choices would strengthen the comparison.
* Baseline Competitiveness: The baselines perform quite comparably to UBER across most environments, with CartPole as a notable exception. This figure deserves a more extensive discussion, as it is currently addressed with only a single sentence at the bottom of page 10.

### Figure 5
* Legend Clarity: The symbols used are not explained in the legend, leaving the black cross unexplained. While this becomes somewhat clearer from Figure 6, each figure should ideally be interpretable independently.
* Comparative Performance: The figure suggests that RS might achieve better performance than UBER with smaller buffer sizes. Could you clarify if this interpretation is correct?

### Figure 6
* Comparison with RS: While the legend is clear, the benefits of UBER over RS are not apparent in this figure. Additional context on UBER’s comparative advantages would strengthen the analysis.

### Figure 7
* RS vs. UBER Performance: Why does RS outperform UBER in the EASY -> HARD setting? Furthermore, why was RS’s buffer not scaled up to match UBER’s buffer size for a direct comparison?

## Minor comments
* Punctuation: The following sentence on page 10 is missing punctuation: “UBER achieved optimal performance with 1700 examples which is 23% less examples than A-FIFO and just a 4% of the total examples collected by PETs.” A comma after “examples” would improve readability.

---

> ### Author Response · Authors · 2024-11-21
>
> We thank the reviewer for their valuable feedback. We address your comments and questions below.
>
> **Figure 2**
>
> **Q**: Color Accessibility: The colors are not color-blind friendly.
>
> **A**: We will update the figures to use a color-blind friendly (seaborn colorblind palette) set to improve accessibility.
>
> **Q**: RS Baseline: Why was RS not included as a baseline?
>
> **A**: Figure 2 shows a motivation experiment to illustrate the limitations of naive MBRL methods while learning a single task with redundant information. In this specific case, the performance of RS is similar to UBER, as the dynamics are learned quickly, and the buffer size can be chosen to be similar. This can be seen in Figure 3. We will include RS for completeness in the next revision.
>
> ---
>
> **Figure 3**
>
> **Q**: Baseline Hyperparameters: Could you elaborate on baseline hyperparameter choices?
>
> **A**: For each method, we include the performance upper and lower bounds. The starting point is the buffer size obtained from UBER. From there, we chose larger and smaller sizes at fixed increments. The upper bound corresponds to a buffer size beyond which increasing it further does not improve performance. We then gradually reduce the buffer size until the algorithm can no longer solve the task (e.g., in CartPole, this would result in instability where the pole eventually falls). We aim to demonstrate the sensitivity of traditional MBRL methods to buffer size selection.
>
> **Q**: Baseline Competitiveness: Why is UBER not much better than baselines, except for CartPole?
>
> **A**: Indeed, for the other environments, we observe that UBER is slightly better but generally on par with RS, with the notable advantage of not requiring a pre-set buffer size. This highlights an important distinction: RS and FIFO exhibit significant sensitivity to buffer size, whereas UBER mitigates this issue. For further context, please refer to common remark 3 about the motivation experiment. We will expand on this discussion in the next revision.
>
> ---
>
> **Figure 5 and 6**
>
> **Q**: Legend Clarity: Symbols (e.g., black cross) are unexplained.
>
> **A**: We will update the legend to define all symbols.
>
> **Q**: Comparative Performance: Does RS achieve better performance with smaller buffers?
>
> **Q**: UBER vs. RS: The benefits of UBER over RS are unclear.
>
> **A**: Your interpretation is correct, RS performs slightly better than UBER in terms of buffer size. However, UBER does not require a pre-defined buffer size, making it more adaptable across tasks. RS, in contrast, is highly sensitive to the chosen buffer size, which can limit its practicality.
>
> ---
>
> **Figure 7**
>
> **Q**: EASY → HARD: Why does RS outperform UBER in this setting? Why wasn’t RS’s buffer scaled for direct comparison?
>
> **A**: We acknowledge the issues in this figure and will address them in the next revision.
>
> As recommended by reviewer krR9, we plan to convert Figures 5, 6, and 7 into tables to improve readability. This change will address many of the identified issues.
>
> ---
>
> ## Requested Changes:
>
> **Q**: Improve Figures: Add clear legends, define symbols, and use a color-blind friendly palette.
>
> **A**: We appreciate the feedback. We will revise all figures to include clear legends, consistent symbols, and a color-blind friendly palette to ensure accessibility and clarity.
>
> **Q**: Showcase UBER Superiority: Experimental evidence supporting UBER’s advantages over RS is limited.
>
> **A**: While RS performs slightly better than UBER in terms of replay buffer size, UBER’s key advantage lies in not requiring a predefined buffer size, which RS does. This flexibility is particularly beneficial in dynamic or multi-task settings. Expanding the experiments to showcase UBER's strengths in such scenarios would require transitioning to more modern methods (e.g., TD-MPCs), which is beyond the scope of this work. However, this does not detract from our core contribution: a method that automatically optimizes the replay buffer without requiring pre-set sizes. See also the common remark 2.
>
> **Q**: Add New Experiment: Could you show where UBER outperforms RS in buffer-constrained or complex tasks?
>
> **A**: This is a limitation of our current method. To address it, we would need to extend our approach (e.g., by moving away from PETS and adopting modern methods like TD-MPCs). However, we emphasize that our main contribution is a method capable of automatically storing examples in the replay buffer without requiring a predefined buffer size.
>
> **Q**: Pixel-Based Experiment: Consider adding a pixel-based experiment.
>
> **A**: Thank you for the suggestion. Adding a pixel-based setting would be a valuable extension. However, PETS does not support pixel-based environments as it requires explicit trajectory projection and dynamics modeling. Exploring pixel-based scenarios is part of our planned follow-up work, but it is currently outside the scope of this paper.
>
> ---
>
> Please do not hesitate to let us know if you have any additional comments.

---

> > ### Comment · Reviewer_beJY · 2024-12-11
> > **Clarification**
> >
> > Thank you for your response.
> >
> > Could you elaborate more on your answer to `Why does RS outperform UBER in the EASY -> HARD setting in figure 7?`
> >
> > UBER performs much worst than RS and is roughly on par with FIFO. I think this requires much more explanation than what is currently in the text. Is there a fundamental limitation of UBER in the case where the target domain (HARD in this case) is out of distribution? I think this is one of the most important experiment of the paper i.e., generalization for a domain other than Masspoint.
> >
> > For completeness, the QA I am referring to:
> > ```
> > Figure 7
> >
> > Q: EASY → HARD: Why does RS outperform UBER in this setting? Why wasn’t RS’s buffer scaled for direct comparison?
> >
> > A: We acknowledge the issues in this figure and will address them in the next revision.
> > ```
> >
> > and the passage in the text:
> >
> > ```
> > Generalization to Unseen Complex Tasks. Figure 7 (left) shows the generalization from EASY to
> > unseen HARD tasks for each method with different hyperparameters. No method generalizes well to hard
> > tasks, with each showing a significant drop in performance compared to PETs at convergence. However,
> > increased experience leads to slightly better generalization.
> > ```

---

> > > ### Comment · Action_Editor_uxrC · 2024-12-19
> > >
> > > Hi Authors,
> > >
> > > Can you please respond to the reviewer's follow-up question? We'd like to have your thoughts on the matter before we make a final decision.

---

> > > ### Author Response · Authors · 2024-12-20
> > > **RS performs worse for 1k samples and better at 10k samples. But finding these settings involves various iterations.**
> > >
> > > Our sincere apologies for the delayed response.
> > >
> > > Q: Generalization to Unseen Complex Tasks. Figure 7 (left) shows the generalization from EASY to unseen HARD tasks for each method with different hyperparameters. No method generalizes well to hard tasks, with each showing a significant drop in performance compared to PETs at convergence. However, increased experience leads to slightly better generalization.
> > >
> > > A: In the very low sample regime of the EASY→HARD setting, RS performs worse than UBER. For 1k samples, UBER achieves 0.4, while RS falls below the chart's range (<0.2). This is not clearly visible in the image, we will correct it.
> > >
> > > We believe there are two possible explanations for why UBER performs worse than RS at 10k samples:
> > >
> > > a.) Selective experience retention: UBER's core design focuses on retaining experiences most useful for the current task, making it less robust when the target task is out-of-distribution. This selective retention works well for within-distribution tasks but may struggle when transitioning to significantly different tasks.
> > >
> > > In contrast, RS performs better in this specific setting due to its diverse experience buffer, which is beneficial when dealing with a small number of tasks. However, RS does not scale as well to scenarios involving more tasks, since the buffer size needs to be carefully tuned.
> > >
> > > These results suggest that achieving strong generalization may require training on a broader set of diverse tasks.
> > >
> > > b.) Variability in the results: All methods, including RS, show low performance in this setting, with significant variability in the results. Additional runs could help clarify these fluctuations. Notably, in the Masspoint setting, UBER demonstrates strong generalization, highlighting its effectiveness when tasks are within the same distribution.
> > >
> > > Q: For completeness, the QA I am referring to: Figure 7
> > >
> > > A: In the bike model for RS, we first ran experiments in the all→all setting, starting with a large buffer and progressively reducing the buffer size until we see a performance drop. The rightmost points in the chart represent this threshold. To show the impact of buffer size on performance, we then selected a smaller buffer to show this drop. Therefore, the chart includes two different points reflecting different buffer configurations.
> > >
> > > We then reported the metrics in the other settings using this selected buffer size. This highlights the inconvenience of having to tune the buffer size in RS, which can be a drawback when compared to UBER.

---

> > > > ### Comment · Action_Editor_uxrC · 2024-12-22
> > > >
> > > > Reviewer beJY,
> > > >
> > > > The authors have responded to your questions. Can you please take a look and, if possible, submit your official recommendation so we can wrap up this review process?

---

### Author Response · Authors · 2024-11-21
**To all reviewers**

We thank all reviewers for their thoughtful comments and constructive feedback. We have addressed your individual questions under their corresponding review. However, we would like to address below the three common questions raised by multiple reviewers. We remain open to discussion, further feedback or clarifications as needed.


1. **Additional environments**
Expanding the experiments to high-dimensional or additional multi-task settings, such as MetaWorld, is outside the scope of the current study. UBER is implemented on top of PETs, which relies on explicitly modeling the environment dynamics incorporating domain knowledge into the dynamics model,  akin to physics-informed Reinforcement Learning  (Appendix 13 shows the equations governing the Mass Point and Bike environments). Therefore, both PETS and UBER can be applied to environments where such dynamics can be readily defined and efficiently learned. Implementing UBER into algorithms such as TD-MPC2 or Dreamer would facilitate the expansion to high dimensional visual inputs, and is a worthy research direction for future work. Instead, in our current work we have developed two environments enabling the definition of continual learning scenarios with tasks that resemble such physics-informed reinforcement learning objectives. We believe that this in itself is a valuable contribution in addition to the experience replay management strategy (UBER).

2. **RS vs. UBER**
RS and UBER are inherently different, and each has their own strengths. UBER addresses a problem that cannot be directly addressed with RS, namely the continuous interaction and learning in the environment when we have no information about the cap size of the experience replay buffer. In all our experiments, the buffer size of RS has been chosen after solving the task with UBER, with the information about the maximum size of buffer UBER used, and choosing partially smaller sizes for comparison. Perhaps this could be explained more clearly. UBER’s key advantage lies in not requiring a predefined buffer size. This flexibility is particularly beneficial in dynamic, never-ending or multi-task settings, where pre-determining the optimal buffer size is challenging.  Our core contribution is a method that automatically optimizes the replay buffer without requiring preset sizes.

3. **Motivation experiment**
The experiment in Section 4 is a preliminary overview on the buffer size growth issue, and not yet defined within the continual learning context. The experiment is in the context of a single task, and aims to highlight the effects of redundancy in experiences accumulated over longer training periods. UBER being included  in Figure 2 is for efficiency, so that it can be also referenced in Section 5, after introducing the algorithm. We believe this might be a source of confusion. In order to improve it, we suggest moving Section 4 to Section 5.1. This will concentrate the single task learning in long sessions (comparing the baselines, UBER and RS in Figure 2). This will displace Section 5.1 to 5.2 and  focus on the buffer efficiency when retraining from scratch using the accumulated buffer (Figure 3) adapted to highlight the relation between discard and buffer size (max discard performance with min buffer size is ideal).  With these changes, the new Section 4 will describe UBER, Section 4.1 the single task learning experiment, and Section 4.2 learning from buffered experiences. The subsequent sections introduce the continual learning setting (Section 6 -> Section 5) and experiments (Section 7 -> Section 6).

We believe the revisions we have provided address the points raised by the reviewers, and we are happy to provide further clarification if needed. Thanks for your time and consideration in reviewing our work.

---

### Decision · Action_Editor_uxrC · 2025-01-16

**Recommendation:** Accept with minor revision

**Comment:**

The paper is an exploratory step in confronting one of many challenges raised in the continual learning setting. There is certainly an audience for these findings and I think the central claim that UBER can selectively retain relevant data when the environment is non-stationary and the length of interaction is indeterminate is sufficiently supported. Many possible clarity revisions were suggested during the review process and I encourage the authors to take them into account in their revision. The area that I would particularly like to look over once more before publication is the discussion of the comparison between UBER and RS; as-is the plots raise a number of questions in an attentive reader and it's important that the paper take a measured, well-supported approach to situating UBER in the context of these existing methods.

**Audience:**

The reviewers all agree that this paper is addressing an important and relevant problem and that progress in this direction would be of interest to members of the TMLR readership.

**Claims And Evidence:**

Overall, the reviewers appreciated the empirical evaluation of the paper's claims. They found that the experiments were well-designed, properly motivated, and targeted at sensible questions.

One concern was raised in the comparison between Reservoir Sampling (RS) and the paper's novel approach (UBER). One reviewer noted that there was insufficient evidence to claim superiority of UBER over RS. Others noted this comparison as well, wanting more analysis from the paper of the comparative strengths and weaknesses. From my own read, I don't think the paper makes the claim that UBER is superior to RS and does explicitly (though briefly) acknowledge that RS performs well in comparison, but has the downside of requiring a fixed, pre-tuned buffer size, which is a deal breaker in the targeted setting of continual learning. So in these experiments, RS is given an advantage over UBER (namely a tuned buffer size suitable for good performance in this particular experiment) making the comparison complicated; a conclusion that one or the other is "better" is not really possible. I do agree with the reviewers that, because this relationship is complicated, these issues deserve more clarity in the text. I also like the idea of replacing or augmenting figures 5, 6, and 7 with tables because the figures are somewhat complicated to interpret.

The other generally shared concern is that the experiments have relatively limited scope and thus don't permit strong statements about UBER's capabilities in more complicated problems. I agree with this, but this is typical of a first step in a new direction. I found the paper's claims to be sufficiently modest and measured to align with the limited scope of the available evidence.